# LIQUID STRUCTURAL STATE-SPACE MODELS

**Ramin Hasani** [†*]
CSAIL, MIT

**Mathias Lechner** [†]
CSAIL, MIT

**Tsun-Hsuan Wang**
CSAIL, MIT

**Makram Chahine**
CSAIL, MIT

**Alexander Amini**
CSAIL, MIT

**Daniela Rus**
CSAIL, MIT

## ABSTRACT

A proper parametrization of state transition matrices of linear state-space models (SSMs) followed by standard nonlinearities enables them to efficiently learn representations from sequential data, establishing the state-of-the-art on an extensive series of long-range sequence modeling benchmarks. In this paper, we show that we can improve further when the structured SSM, such as S4, is given by a linear liquid time-constant (LTC) state-space model. LTC neural networks are causal continuous-time neural networks with an input-dependent state transition module, which makes them learn to adapt to incoming inputs at inference. We show that by using a diagonal plus low-rank decomposition of the state transition matrix introduced in S4, and a few simplifications, the LTC-based structured state-space model, dubbed Liquid-S4, improves generalization across sequence modeling tasks with long-term dependencies such as image, text, audio, and medical time-series, with an average performance of 87.32% on the Long-Range Arena benchmark. On the full raw Speech Command recognition dataset, Liquid-S4 achieves 96.78% accuracy with a 30% reduction in parameter counts compared to S4. The additional gain in performance is the direct result of the Liquid-S4's kernel structure that takes into account the similarities of the input sequence samples during training and inference.

## 1 INTRODUCTION

Learning representations from sequences of data requires expressive temporal and structural credit assignment. In this space, the continuous-time neural network class of liquid time-constant networks (LTC) (Hasani et al., 2021b) has shown theoretical and empirical evidence for their expressivity and their ability to capture the cause and effect of a given task from high-dimensional sequential demonstrations (Lechner et al., 2020a; Vorbach et al., 2021; Wang et al., 2022; Hasani et al., 2022; Yin et al., 2022). Liquid networks are nonlinear state-space models (SSMs) with an input-dependent state transition module that enables them to learn to adapt the dynamics of the model to incoming inputs, at inference, as they are dynamic causal models (Friston et al., 2003). Their complexity, however, is bottlenecked by their differential equation numerical solver that limits their scalability to longer-term sequences. How can we take advantage of LTC's generalization and causality capabilities and scale them to competitively learn long-range sequences without gradient issues, compared to advanced recurrent neural networks (RNNs) (Rusch & Mishra, 2021a; Erichson et al., 2021; Gu et al., 2020a), convolutional networks (CNNs) (Lea et al., 2016; Romero et al., 2021b; Cheng et al., 2022), and attention-based models (Vaswani et al., 2017)?

In this work, we set out to leverage the elegant formulation of structured state-space models (S4) (Gu et al., 2022a) to obtain linear liquid network instances that possess the approximation capabilities of both S4 and LTCs. This is because structured SSMs are shown to largely dominate advanced RNNs, CNNs, and Transformers across many data modalities such as text, sequence of pixels, audio, and time series (Gu et al., 2021; 2022a;b; Gupta, 2022). structured SSMs achieve such impressive performance by using three main mechanisms: 1) High-order polynomial projection operators (HiPPO)

---

[*]correspondence to: `rhasani@mit.edu`  Code: `https://github.com/raminmh/liquid-s4`
[†] indicates authors with equal contributions.

(Gu et al., 2020a) that are applied to state and input transition matrices to memorize signals' history, 2) diagonal plus low-rank parametrization of the obtained HiPPO (Gu et al., 2022a), and 3) an efficient (convolution) kernel computation of an SSM's transition matrices in the frequency domain, transformed back in time via an inverse Fourier transformation (Gu et al., 2022a).

To combine S4 and LTCs, instead of modeling sequences by linear state-space models of the form $\dot{x} = \mathbf{A}\,x + \mathbf{B}\,u$, $y = \mathbf{C}\,x$, (as done in structured and diagonal SSMs (Gu et al., 2022a;b)), we propose to use a linearized LTC state-space model (Hasani et al., 2021b), given by the following dynamics: $\dot{x} = (\mathbf{A} + \mathbf{B}\,u)\,x + \mathbf{B}\,u$, $y = \mathbf{C}\,x$. We show that this dynamical system can also be efficiently solved via the same parametrization of S4, giving rise to an additional convolutional Kernel that accounts for the similarities of lagged signals. We call the obtained model Liquid-S4. Through extensive empirical evaluation, we show that Liquid-S4 consistently leads to better generalization performance compared to all variants of S4, CNNs, RNNs, and Transformers across many time-series modeling tasks. In particular, we achieve SOTA performance on the Long Range Arena benchmark (Tay et al., 2020b). To sum up, we make the following contributions:

1. We introduce Liquid-S4, a new state-space model that encapsulates the generalization and causality capabilities of liquid networks as well as the memorization and scalability of S4.
2. We achieve state-of-the-art performance on pixel-level sequence classification, text, speech recognition, and all six tasks of the long-range arena benchmark with an average accuracy of 87.32%. On the full raw Speech Command recognition dataset, Liquid-S4 achieves 96.78% accuracy with a 30% reduction in parameters. Finally, on the BIDMC vital signs dataset, Liquid-S4 achieves SOTA in all modes.

## 2 RELATED WORKS

**Learning Long-Range Dependencies with RNNs.** Sequence modeling can be performed autoregressively with RNNs which possess persistent states (Little, 1974) originated from Ising (Brush, 1967) and Hopfield networks (Hopfield, 1982; Ramsauer et al., 2020). Discrete RNNs approximate continuous dynamics step-by-steps via dependencies on the history of their hidden states, and continuous-time (CT) RNNs use ordinary differential equation (ODE) solvers to unroll their dynamics with more elaborate temporal steps (Funahashi & Nakamura, 1993).

CT-RNNs can perform remarkable credit assignment in sequence modeling problems both on regularly sampled, irregularly-sampled data (Pearson et al., 2003; Li & Marlin, 2016; Belletti et al., 2016; Roy & Yan, 2020; Foster, 1996; Amigó et al., 2012; Kowal et al., 2019), by turning the spatiotemproal dependencies into vector fields (Chen et al., 2018), enabling better generalization and expressivity (Massaroli et al., 2020; Hasani et al., 2021b). Numerous works have studied their characteristics to understand their applicability and limitations in learning sequential data and flows (Lechner et al., 2019; Dupont et al., 2019; Durkan et al., 2019; Jia & Benson, 2019; Grunbacher et al., 2021; Hanshu et al., 2020; Holl et al., 2020; Quaglino et al., 2020; Kidger et al., 2020; Hasani et al., 2020; Liebenwein et al., 2021; Gruenbacher et al., 2022).

However, when these RNNs are trained by gradient descent (Rumelhart et al., 1986; Allen-Zhu & Li, 2019; Sherstinsky, 2020), they suffer from the vanishing/exploding gradients problem, which makes difficult the learning of long-term dependencies in sequences (Hochreiter, 1991; Bengio et al., 1994). This issue happens in both discrete RNNs such as GRU-D with its continuous delay mechanism (Che et al., 2018) and Phased-LSTMs (Neil et al., 2016), and continuous RNNs such as ODE-RNNs (Rubanova et al., 2019), GRU-ODE (De Brouwer et al., 2019), Log-ODE methods (Morrill et al., 2020) which compresses the input time-series by time-continuous path signatures (Friz & Victoir, 2010), and neural controlled differential equations (Kidger et al., 2020), and liquid time-constant networks (LTCs) (Hasani et al., 2021b).

Numerous solutions have been proposed to resolve these gradient issues to enable long-range dependency learning. Examples include discrete gating mechanisms in LSTMs (Hochreiter & Schmidhuber, 1997; Greff et al., 2016; Hasani et al., 2019), GRUs (Chung et al., 2014), continuous gating mechanisms such as CfCs (Hasani et al., 2021a), hawks LSTMs (Mei & Eisner, 2017), IndRNNs (Li et al., 2018), state regularization (Wang & Niepert, 2019), unitary RNNs (Jing et al., 2019), dilated RNNs (Chang et al., 2017), long memory stochastic processes (Greaves-Tunnell & Harchaoui, 2019), recurrent kernel networks (Chen et al., 2019), Lipschitz RNNs (Erichson et al., 2021), symmetric skew decomposition (Wisdom et al., 2016), infinitely many updates in iRNNs

(Kag et al., 2019), coupled oscillatory RNNs (coRNNs) (Rusch & Mishra, 2021a), mixed-memory RNNs (Lechner & Hasani, 2021), and Legendre Memory Units (Voelker et al., 2019).

**Learning Long-range Dependencies with CNNs and Transformers.** RNNs are not the only solution to learning long-range dependencies. Continuous convolutional kernels such as CKConv (Romero et al., 2021b) and (Romero et al., 2021a), and circular dilated CNNs (Cheng et al., 2022) have shown to be efficient in modeling long sequences faster than RNNs. There has also been a large series of works showing the effectiveness of attention-based methods for modeling spatiotemporal data. A large list of these models is listed in Table 6. These baselines have recently been largely outperformed by the structured state-space models (Gu et al., 2022a).

**State-Space Models.** SSMs are well-established frameworks to study deterministic and stochastic dynamical systems (Kalman, 1960). Their state and input transition matrices can be directly learned by gradient descent to model sequences of observations (Lechner et al., 2020b; Hasani et al., 2021b; Gu et al., 2021). In a seminal work, Gu et al. (2022a) showed that with a couple of fundamental algorithmic methods on memorization and computation of input sequences, SSMs can turn into the most powerful sequence modeling framework to-date, outperforming advanced RNNs, temporal and continuous CNNs (Cheng et al., 2022; Romero et al., 2021b;a) and a wide variety of Transformers (Vaswani et al., 2017), available in Table 6 by a significant margin.

The key to their numerical performance is their derivation of higher-order polynomial projection (HiPPO) matrix (Gu et al., 2020a) obtained by a scaled Legendre measure (LegS) inspired by the Legendre Memory Units (Voelker et al., 2019) to memorize input sequences. Their efficient runtime and memory are derived from their normal plus-low rank representation. It was also shown recently that diagonal SSMs (S4D) (Gupta, 2022) could be as performant as S4 in learning long sequences when parametrized and initialized properly (Gu et al., 2022b;c). Concurrent with our work, there is also a new variant of S4 introduced as simplified-S4 (S5) (Smith et al., 2022) that tensorizes the 1-D operations of S4 to gain a more straightforward realization of SSMs. Here, we introduce Liquid-S4, which is obtained by a more expressive SSM, namely liquid time-constant (LTC) representation (Hasani et al., 2021b) which achieves SOTA performance across many benchmarks.

## 3 SETUP AND METHODOLOGY

In this section, we first revisit the necessary background to formulate our liquid structured state-space models. We then set up and sketch our technical contributions.

### 3.1 BACKGROUND: STRUCTURED STATE-SPACE MODELS (S4)

We aim to design an end-to-end sequence modeling framework built by SSMs. A continuous-time SSM representation of a linear dynamical system is given by the following set of equations:

$$\dot{\boldsymbol{x}}(t) = \mathbf{A}\,\boldsymbol{x}(t) + \mathbf{B}\,u(t), \quad y(t) = \mathbf{C}\,\boldsymbol{x}(t) + \mathbf{D}\,u(t). \tag{1}$$

Here, $\boldsymbol{x}(t)$ is an $N$-dimensional latent state, receiving a 1-dimensional input signal $u(t)$, and computing a 1-dimensional output signal $y(t)$. $\mathbf{A}^{(N \times N)}$, $\mathbf{B}^{(N \times 1)}$, $\mathbf{C}^{(1 \times N)}$ and $\mathbf{D}^{(1 \times 1)}$ are system's parameters. For the sake of brevity, throughout our analysis, we set $\mathbf{D} = 0$ as it can be added eventually after construction of our main results in the form of a skip connection (Gu et al., 2022a).

**Discretization of SSMs.** In order to create a sequence-to-sequence model similar to a recurrent neural network (RNN), we discretize the continuous-time representation of SSMs by the trapezoidal rule (bilinear transform) as follows (sampling step = $\delta t$) (Gu et al., 2022a):

$$x_k = \overline{\mathbf{A}}\,x_{k-1} + \overline{\mathbf{B}}\,u_k, \quad y_k = \overline{\mathbf{C}}\,x_k \tag{2}$$

This is obtained via the following modifications to the transition matrices:

$$\overline{\mathbf{A}} = (\mathbf{I} - \frac{\delta t}{2}\mathbf{A})^{-1}(\mathbf{I} + \frac{\delta t}{2}\mathbf{A}), \quad \overline{\mathbf{B}} = (\mathbf{I} - \frac{\delta t}{2}\mathbf{A})^{-1}\,\delta t\,\mathbf{B}, \quad \overline{\mathbf{C}} = \mathbf{C} \tag{3}$$

With this transformation, we constructed a discretized seq-2-seq model that can map the input $u_k$ to output $y_k$, via the *hidden* state $x_k \in \mathbb{R}^N$. $\overline{\mathbf{A}}$ is the hidden transition matrix, $\overline{\mathbf{B}}$ and $\overline{\mathbf{C}}$ are input and output transition matrices, respectively.

**Creating a Convolutional Representation of SSMs.** The system described by Eq. 2 and Eq. 3, can be trained via gradient descent to learn to model sequences, in a sequential manner which is not scalable. To improve this, we can write the discretized SSM in Eq. 2 as a discrete convolutional kernel. To construct the convolutional kernel, let us unroll the system of Eq. 2 in time as follows, assuming a zero initial hidden states $x_{-1} = 0$:

$$x_0 = \overline{\mathbf{B}} u_0, \qquad x_1 = \overline{\mathbf{AB}} u_0 + \overline{\mathbf{B}} u_1, \qquad x_2 = \overline{\mathbf{A}}^2 \overline{\mathbf{B}} u_0 + \overline{\mathbf{AB}} u_1 + \overline{\mathbf{B}} u_2, \qquad \dots \qquad (4)$$

$$y_0 = \overline{\mathbf{CB}} u_0, \quad y_1 = \overline{\mathbf{CAB}} u_0 + \overline{\mathbf{CB}} u_1, \quad y_2 = \overline{\mathbf{CA}}^2 \overline{\mathbf{B}} u_0 + \overline{\mathbf{CAB}} u_1 + \overline{\mathbf{CB}} u_2, \quad \dots$$

The mapping $u_{0,k} \to y_k$ can now be formulated into a convolutional kernel explicitly:

$$y_k = \overline{\mathbf{CA}}^k \overline{\mathbf{B}} u_0 + \overline{\mathbf{CA}}^{k-1} \overline{\mathbf{B}} u_1 + \dots \overline{\mathbf{CAB}} u_{k-1} + \overline{\mathbf{CB}} u_k, \qquad y = \overline{\mathbf{K}} * u \qquad (5)$$

$$\overline{\mathbf{K}} \in \mathbb{R}^L := \mathcal{K}_L(\overline{\mathbf{C}}, \overline{\mathbf{A}}, \overline{\mathbf{B}}) := \left( \overline{\mathbf{CA}}^i \overline{\mathbf{B}} \right)_{i \in [L]} = \left( \overline{\mathbf{CB}}, \overline{\mathbf{CAB}}, \dots, \overline{\mathbf{CA}}^{L-1} \overline{\mathbf{B}} \right) \qquad (6)$$

Eq. 5 is a non-circular convolutional kernel. Gu et al. (2022a) showed that under the condition that $\overline{\mathbf{K}}$ is known, it can be solved very efficiently by a black-box Cauchy kernel computation pipeline.

**Computing S4 Kernel Efficiently:** Gu et al. (2022a) showed that the S4 convolution kernel could be computed efficiently using the following elegant parameterization tricks:

- To obtain better representations in sequence modeling schemes by SSMs, instead of randomly initializing the transition matrix $\mathbf{A}$, we can use the normal plus low-Rank (NPLR) matrix below, called the Hippo Matrix (Gu et al., 2020a) which is obtained by the scaled Legendre measure (LegS) (Gu et al., 2021; 2022a):

$$\textbf{(HiPPO Matrix)} \qquad \mathbf{A}_{nk} = - \begin{cases} (2n+1)^{1/2}(2k+1)^{1/2} & \text{if } n > k \\ n+1 & \text{if } n = k \\ 0 & \text{if } n < k \end{cases} \qquad (7)$$

- The NPLR representation of this matrix is the following (Gu et al., 2022a):

$$\mathbf{A} = \mathbf{V} \mathbf{\Lambda} \mathbf{V}^* - \mathbf{P} \mathbf{Q}^\top = \mathbf{V} \left( \mathbf{\Lambda} - (\mathbf{V}^* \mathbf{P})(\mathbf{V}^* \mathbf{Q})^* \right) \mathbf{V}^* \qquad (8)$$

Here, $\mathbf{V} \in \mathbb{C}^{N \times N}$ is a unitary matrix, $\mathbf{\Lambda}$ is diagonal, and $\mathbf{P}, \mathbf{Q} \in \mathbb{R}^{N \times r}$ are the low-rank factorization. Eq. 7 is normal plus low rank with r = 1 (Gu et al., 2022a). With the decomposition in Eq. 8, we can obtain $\mathbf{A}$ over complex numbers in the form of diagonal plus low-rank (DPLR) (Gu et al., 2022a).

- Vectors $B_n$ and $P_n$ are initialized by $B_n = (2n+1)^{\frac{1}{2}}$ and $P_n = (n+1/2)^{\frac{1}{2}}$ (Gu et al., 2022b). Both vectors are trainable.

- Furthermore, it was shown in Gu et al. (2022b) that with Eq. 8, the eigenvalues of $\mathbf{A}$ might be on the right half of the complex plane, resulting in numerical instability. To resolve this, Gu et al. (2022b) recently proposed to use the parametrization $\mathbf{\Lambda} - \mathbf{P}\mathbf{P}^*$ instead of $\mathbf{\Lambda} - \mathbf{P}\mathbf{Q}^*$.

- Computing the powers of $\mathbf{A}$ in direct calculation of the S4 kernel $\overline{\mathbf{K}}$ is computationally expensive. S4 computes the spectrum of $\overline{\mathbf{K}}$ instead of direct computations, which reduces the problem of matrix powers to matrix inverse computation (Gu et al., 2022a). S4 then computes this convolution kernel via a black-box Cauchy Kernel efficiently, and recovers $\overline{\mathbf{K}}$ by an inverse Fourier Transform (iFFT) (Gu et al., 2022a).

## 3.2 Liquid Structural State-Space Models

In this work, we construct a convolutional kernel corresponding to a linearized version of LTCs (Hasani et al., 2021b); an expressive class of continuous-time neural networks that demonstrate attractive generalizability out-of-distribution and are dynamic causal models (Vorbach et al., 2021; Friston et al., 2003; Hasani et al., 2020). In their general form, the state of a liquid time-constant network at each time-step is given by the set of ODEs described below (Hasani et al., 2021b):

$$\frac{d\boldsymbol{x}(t)}{dt} = - \underbrace{\left[ \boldsymbol{A} + \boldsymbol{B} \odot f(\boldsymbol{x}(t), \boldsymbol{u}(t), t, \theta) \right]}_{\text{Liquid time-constant}} \odot \boldsymbol{x}(t) + \boldsymbol{B} \odot f(\boldsymbol{x}(t), \boldsymbol{u}(t), t, \theta). \qquad (9)$$

In this expression, $x^{(N\times 1)}(t)$ is the vector of hidden state of size $N$, $u^{(m\times 1)}(t)$ is an input signal with $m$ features, $\boldsymbol{A}^{(N\times 1)}$ is a time-constant state-transition mechanism, $\boldsymbol{B}^{(N\times 1)}$ is a bias vector, and $\odot$ represents the Hadamard product. $f(.)$ is a bounded nonlinearity parametrized by $\theta$.

Our objective is to show how the liquid time-constant (i.e., an input-dependent state transition mechanism in state-space models can enhance its generalization capabilities by accounting for the covariance of the input samples. To do this, we linearize the LTC formulation of Eq. 9 in the following to better connect the model to SSMs. Let's dive in:

**Linear Liquid Time-Constant State-Space Model.** A Linear LTC SSM can be presented by the following coupled bilinear (first order bilinear Taylor approximation (Penny et al., 2005)) equation:

$$\dot{\boldsymbol{x}}(t) = \left[\mathbf{A} + \mathbb{I}_N\mathbf{B}\,u(t)\right]\boldsymbol{x}(t) + \mathbf{B}\,u(t), \qquad y(t) = \mathbf{C}\,\boldsymbol{x}(t) \tag{10}$$

Similar to Eq. 1, $x(t)$ is an $N$-dimensional latent state, receiving a 1-dimensional input signal $u(t)$, and computing a 1-dimensional output signal $y(t)$. $\mathbf{A}^{(N\times N)}$, $\mathbf{B}^{(N\times 1)}$, and $\mathbf{C}^{(1\times N)}$. Note that $\mathbf{D}$ is set to zero for simplicity. In Eq. 10, $\mathbb{J}_N$ is an $N \times N$ unit matrix that adds $\mathbf{B}\,u(t)$ element-wise to $\mathbf{A}$. This dynamical system allows the coefficient (state transition compartment) of state vector $x(t)$ to be input dependent which, as a result, allows us to realize more complex dynamics.

**Discretization of Liquid-SSMs.** We can use a forward Euler transformation to discretize Eq. 10 into the following discretization:

$$x_k = \left(\overline{\mathbf{A}} + \overline{\mathbf{B}}\,u_k\right)x_{k-1} + \overline{\mathbf{B}}\,u_k, \qquad y_k = \overline{\mathbf{C}}\,x_k \tag{11}$$

The discretized parameters would then correspond to: $\overline{\mathbf{A}} = \mathbf{I} + \frac{\delta t}{2}\mathbf{A}$, $\overline{\mathbf{B}} = \delta t\,\mathbf{B}$, and $\overline{\mathbf{C}} = \mathbf{C}$, which are function of the continuous-time coefficients $\mathbf{A}$, $\mathbf{B}$, and $\mathbf{C}$, and the discretization step $\delta t$. Given the properties of the transition matrices $\mathbf{A}$ and $\mathbf{B}$, and ranges of $\delta t$, we could use the more stable bilinear discretization of matrices $\mathbf{A}$ and $\mathbf{B}$, of Eq. 3 as well, as the Forward Euler discretization and the bilinear transformation of $\mathbf{A}$ and $\mathbf{B}$ presented in Eq. 3 stay close to each other (Appendix D).

**Creating a Convolutional Representation of Liquid-SSMs.** Similar to Eq. 4, we first unroll the Liquid-SSM in time to construct a convolutional kernel of it. By assuming $x_{-1} = 0$, we have:

$$
\begin{aligned}
x_0 &= \overline{\mathbf{B}}u_0, \quad y_0 = \overline{\mathbf{CB}}u_0 \\
x_1 &= \overline{\mathbf{AB}}u_0 + \overline{\mathbf{B}}u_1 + \overline{\mathbf{B}}^2 u_0 u_1, \quad y_1 = \overline{\mathbf{CAB}}u_0 + \overline{\mathbf{CB}}u_1 + \overline{\mathbf{CB}}^2 u_0 u_1 \\
x_2 &= \overline{\mathbf{A}}^2\overline{\mathbf{B}}u_0 + \overline{\mathbf{AB}}u_1 + \overline{\mathbf{B}}u_2 + \overline{\mathbf{AB}}^2 u_0 u_1 + \overline{\mathbf{AB}}^2 u_0 u_2 + \overline{\mathbf{B}}^2 u_1 u_2 + \overline{\mathbf{B}}^3 u_0 u_1 u_2 \\
y_2 &= \overline{\mathbf{CA}}^2\overline{\mathbf{B}}u_0 + \overline{\mathbf{CAB}}u_1 + \overline{\mathbf{CB}}u_2 + \overline{\mathbf{CAB}}^2 u_0 u_1 + \overline{\mathbf{CAB}}^2 u_0 u_2 + \overline{\mathbf{CB}}^2 u_1 u_2 + \overline{\mathbf{CB}}^3 u_0 u_1 u_2, \quad \dots
\end{aligned}
\tag{12}
$$

The resulting expressions of the Liquid-SSM at each time step consist of two types of weight configurations: 1. Weights corresponding to the mapping of individual time instances of inputs independently, shown in black in Eq. 12, and 2. Weights associated with all orders of auto-correlation of the input signal, shown in violet in Eq. 12. The first set of weights corresponds to the convolutional kernel of the simple SSM, shown by Eq. 5 and Eq. 6, whereas the second set leads to the design of an additional input correlation kernel, which we call the *liquid* kernel. These kernels generate the following input-output mapping:

$$
\begin{aligned}
y_k = {} & \overline{\mathbf{CA}}^k\overline{\mathbf{B}}u_0 + \overline{\mathbf{CA}}^{k-1}\overline{\mathbf{B}}u_1 + \dots \overline{\mathbf{CAB}}u_{k-1} + \overline{\mathbf{CB}}u_k + \\
& \sum_{p=2}^{\mathcal{P}} \sum_{u_i u_{i+1}\,\dots\,u_p\in\Pi(k+1,p)} \overline{\mathbf{CA}}^{(k+1-p-i)}\overline{\mathbf{B}}^p u_i u_{i+1}\,\dots\,u_p \\
& \text{for } i\in\mathbb{Z}\text{ and }i\geq 0, \quad \rightarrow \quad y = \overline{\mathbf{K}} * u + \overline{\mathbf{K}}_{\text{liquid}} * u_{\text{correlations}}.
\end{aligned}
\tag{13}
$$

Here, $\Pi(k+1, p)$ represents $\binom{k+1}{p}$ permuted indices. For instance, let us assume we have a 1-dimensional input signal $u(t)$ of length $L = 100$ on which we run the liquid-SSM kernel. We set the hyperparameters $\mathcal{P} = 4$. This value represents the maximum order of the correlation terms we

**Algorithm 1** LIQUID-S4 KERNEL - The S4 convolution kernel (highlighted in black) is used from Gu et al. (2022a) and Gu et al. (2022b). Liquid kernel computation is highlighted in purple.

---

**Input:** S4 parameters $\mathbf{\Lambda}, \boldsymbol{P}, \boldsymbol{B}, \boldsymbol{C} \in \mathbb{C}^N$, step size $\Delta$, liquid kernel order $\mathcal{P}$, inputs seq length $L$, liquid kernel sequence length $\tilde{L}$

**Output:** SSM convolution kernel $\overline{\boldsymbol{K}} = \mathcal{K}_L(\overline{\boldsymbol{A}}, \overline{\boldsymbol{B}}, \overline{\boldsymbol{C}})$ and SSM liquid kernel $\overline{\boldsymbol{K}}_{liquid} = \mathcal{K}_{\tilde{L}}(\overline{\boldsymbol{A}}, \overline{\boldsymbol{B}}, \overline{\boldsymbol{C}})$ for $\boldsymbol{A} = \mathbf{\Lambda} - \boldsymbol{P}\boldsymbol{P}^*$ (Eq. 6)

1: $\widetilde{\boldsymbol{C}} \leftarrow \left(\boldsymbol{I} - \overline{\boldsymbol{A}}^L\right)^* \overline{\boldsymbol{C}}$          ▷ Truncate SSM generating function (SSMGF) to length $L$

2: $\begin{bmatrix} k_{00}(\omega) & k_{01}(\omega) \\ k_{10}(\omega) & k_{11}(\omega) \end{bmatrix} \leftarrow \begin{bmatrix} \widetilde{\boldsymbol{C}} & \boldsymbol{P} \end{bmatrix}^* \left(\frac{2}{\Delta}\frac{1-\omega}{1+\omega} - \mathbf{\Lambda}\right)^{-1} [\boldsymbol{B} \; \boldsymbol{P}]$     ▷ Black-box Cauchy kernel

3: $\hat{\boldsymbol{K}}(\omega) \leftarrow \frac{2}{1+\omega}\left[k_{00}(\omega) - k_{01}(\omega)(1 + k_{11}(\omega))^{-1}k_{10}(\omega)\right]$     ▷ Woodbury Identity

4: $\hat{\boldsymbol{K}} = \{\hat{\boldsymbol{K}}(\omega) : \omega = \exp(2\pi i\frac{k}{L})\}$     ▷ Evaluate SSMGF at all roots of unity $\omega \in \Omega_L$

5: $\overline{\boldsymbol{K}} \leftarrow \text{iFFT}(\hat{\boldsymbol{K}})$     ▷ Inverse Fourier Transform

6: **if** Mode == KB **then**     ▷ Liquid-S4 Kernel as shown in Eq. 14

7:     **for** $p$ in $\{2, \ldots, \mathcal{P}\}$ **do**

8:        $\overline{\boldsymbol{K}}_{\text{liquid}=p} = \left[\overline{\boldsymbol{K}}_{(L-\tilde{L},L)} \odot \overline{\boldsymbol{B}}^{p-1}_{(L-\tilde{L},L)}\right] * \mathbf{J}_{\tilde{L}}$     ▷ $\mathbf{J}_{\tilde{L}}$ is a backward identity matrix

9:        $\overline{\boldsymbol{K}}_{\text{liquid}}.\text{append}(\overline{\boldsymbol{K}}_{\text{liquid}=p})$

10:     **end for**

11: **else if** Mode == PB **then**     ▷ Liquid-S4 Kernel of Eq. 14 with $\overline{\boldsymbol{A}}$ reduced to Identity.

12:     **for** $p$ in $\{2, \ldots, \mathcal{P}\}$ **do**

13:        $\overline{\boldsymbol{K}}_{\text{liquid}=p} = \overline{\boldsymbol{C}} \odot \overline{\boldsymbol{B}}^{p-1}_{(L-\tilde{L},L)}$

14:        $\overline{\boldsymbol{K}}_{\text{liquid}}.\text{append}(\overline{\boldsymbol{K}}_{\text{liquid}=p})$

15:     **end for**

16: **end if**

---

would want to take into account to output a decision. This means that the signal $u_{\text{correlations}}$ in Eq. 13 will contain all combinations of 2 order correlation signals $\binom{L+1}{2}$, $u_i u_j$, 3 order $\binom{L+1}{3}$, $u_i u_j u_k$ and 4 order signals $\binom{L+1}{4}$, $u_i u_j u_k u_l$. The kernel weights corresponding to this auto-correlation signal are given in Appendix A. This additional kernel takes the temporal similarities of incoming input samples into consideration. This way, Liquid-SSM gives rise to a more general sequence modeling framework. The liquid convolutional kernel, $\overline{\mathbf{K}}_{\text{liquid}}$ is as follows:

$$\overline{\mathbf{K}}_{\text{liquid}} \in \mathbb{R}^{\tilde{L}} := \mathcal{K}_L(\overline{\mathbf{C}}, \overline{\mathbf{A}}, \overline{\mathbf{B}}) := \left(\overline{\mathbf{C}\mathbf{A}}^{(\tilde{L}-i-p)}\overline{\mathbf{B}}^p\right)_{i\in[\tilde{L}], \; p\in[\mathcal{P}]} = \left(\overline{\mathbf{C}\mathbf{A}}^{\tilde{L}-2}\overline{\mathbf{B}}^2, \ldots, \overline{\mathbf{C}\mathbf{B}}^p\right) \quad (14)$$

**How to compute Liquid-S4 kernel efficiently?** $\overline{\mathbf{K}}_{\text{liquid}}$ possess similar structure to the S4 kernel. In particular, we have:

**Proposition 1.** *The Liquid-S4 kernel for each order $p \in \mathcal{P}$, $\overline{\mathbf{K}}_{liquid}$, can be computed by the anti-diagonal transformation (flip operation) of the product of the S4 convolution kernel, $\overline{\mathbf{K}} = \left(\overline{\mathbf{CB}}, \overline{\mathbf{CAB}}, \ldots, \overline{\mathbf{CA}}^{L-1}\overline{\mathbf{B}}\right)$, and a vector $\overline{\mathbf{B}}^{p-1} \in \mathbb{R}^N$.*

The proof is given in Appendix. Proposition 1 indicates that the Liquid-s4 kernel can be obtained from the precomputed S4 kernel and a Hadamard product of that kernel with the transition vector $\overline{\boldsymbol{B}}$ powered by the chosen liquid order. This is illustrated in Algorithm 1, lines 6 to 10, corresponding to a mode we call KB, which stands for Kernel $\times$ B.

Additionally, we introduce a simplified Liquid-S4 kernel that is easier to compute while is as expressive as or even better performing than the KB kernel. To obtain this, we set the transition matrix $\overline{\boldsymbol{A}}$ in Liquid-S4 of Eq. 14, with an identity matrix, only for the input correlation terms. This way, the Liquid-s4 Kernel for a given liquid order $p \in \mathcal{P}$ reduces to the following expression:

$$(\textbf{Liquid-S4 - PB}) \qquad \overline{\mathbf{K}}_{\text{liquid}=p} \in \mathbb{R}^{\tilde{L}} := \mathcal{K}_L(\overline{\mathbf{C}}, \overline{\mathbf{B}}) := \left(\overline{\mathbf{C}\mathbf{B}}^p\right)_{i\in[\tilde{L}], \; p\in[\mathcal{P}]} \quad (15)$$

We call this kernel Liquid-S4 - PB, as it is obtained by powers of the vector $\overline{\boldsymbol{B}}$. The computational steps to get this kernel is outlined in Algorithm 1 lines 11 to 15.

Table 1: Performance on Long Range Arena Tasks. Numbers indicate validation accuracy (standard deviation). The accuracy of models denoted by * is reported from (Tay et al., 2020b). Methods denoted by ** are reported from (Gu et al., 2022a). The rest of the models' performance results are reported from the cited paper. See Appendix for accuracy on test set.

| Model (input length) | ListOps 2048 | IMDB 2048 | AAN 4000 | CIFAR 1024 | Pathfinder 1024 | Path-X 16384 | Avg. |
|---|---|---|---|---|---|---|---|
| Random* | 10.00 | 50.00 | 50.00 | 10.00 | 50.00 | 50.00 | 36.67 |
| Transformer* (Vaswani et al., 2017) | 36.37 | 64.27 | 57.46 | 42.44 | 71.40 | x | 54.39 |
| Local Att.* (Tay et al., 2020b) | 15.82 | 52.98 | 53.39 | 41.46 | 66.63 | x | 46.06 |
| Sparse Transformer* (Child et al., 2019) | 17.07 | 63.58 | 59.59 | 44.24 | 71.71 | x | 51.24 |
| Longformer* (Beltagy et al., 2020) | 35.63 | 62.85 | 56.89 | 42.22 | 69.71 | x | 53.46 |
| Linformer* (Wang et al., 2020) | 16.13 | 65.90 | 53.09 | 42.34 | 75.30 | x | 50.55 |
| Reformer* (Kitaev et al., 2019) | 37.27 | 56.10 | 53.40 | 38.07 | 68.50 | x | 50.56 |
| Sinkhorn Trans.* (Tay et al., 2020a) | 33.67 | 61.20 | 53.83 | 41.23 | 67.45 | x | 51.23 |
| BigBird* (Zaheer et al., 2020) | 36.05 | 64.02 | 59.29 | 40.83 | 74.87 | x | 55.01 |
| Linear Trans.* (Katharopoulos et al., 2020) | 16.13 | 65.90 | 53.09 | 42.34 | 75.30 | x | 50.46 |
| Performer* (Choromanski et al., 2020) | 18.01 | 65.40 | 53.82 | 42.77 | 77.05 | x | 51.18 |
| FNet** (Lee-Thorp et al., 2021) | 35.33 | 65.11 | 59.61 | 38.67 | 77.80 | x | 54.42 |
| Nyströmformer** (Xiong et al., 2021) | 37.15 | 65.52 | 79.56 | 41.58 | 70.94 | x | 57.46 |
| Luna-256** Ma et al. (2021) | 37.25 | 64.57 | 79.29 | 47.38 | 77.72 | x | 59.37 |
| H-Transformer-1D** (Zhu & Soricut, 2021) | 49.53 | 78.69 | 63.99 | 46.05 | 68.78 | x | 61.41 |
| CDIL (Cheng et al., 2022) | 44.05 | 86.78 | 85.36 | 66.91 | 91.70 | x | 74.96 |
| DSS (Gupta, 2022) | 57.6 | 76.6 | 87.6 | 85.8 | 84.1 | 85.0 | 79.45 |
| S4 (original)** (Gu et al., 2022a) | 58.35 | 76.02 | 87.09 | 87.26 | 86.05 | 88.10 | 80.48 |
| S4-LegS (Gu et al., 2022b) | 59.60 (0.07) | 86.82 (0.13) | 90.90 (0.15) | 88.65 (0.23) | 94.20 (0.25) | 96.35 | 86.09 |
| S4-FouT (Gu et al., 2022b) | 57.88 (1.90) | 86.34 (0.31) | 89.66 (0.88) | 89.07 (0.19) | 94.46 (0.26) | x | 77.90 |
| S4-LegS/FouT (Gu et al., 2022c) | 60.45 (0.75) | 86.78 (0.26) | 90.30 (0.28) | 89.00 (0.26) | 94.44 (0.08) | x | 78.50 |
| S4D-LegS (Gu et al., 2022b) | 60.47 (0.34) | 86.18 (0.43) | 89.46 (0.14) | 88.19 (0.26) | 93.06 (1.24) | 91.95 | 84.89 |
| S4D-Inv (Gu et al., 2022b) | 60.18 (0.35) | 87.34 (0.20) | 91.09 (0.01) | 87.83 (0.37) | 93.78 (0.25) | 92.80 | 85.50 |
| S4D-Lin (Gu et al., 2022b) | 60.52 (0.51) | 86.97 (0.23) | 90.96 (0.09) | 87.93 (0.34) | 93.96 (0.60) | x | 78.39 |
| S5 original (Smith et al., 2022) | 61.00 | 86.51 | 88.26 | 86.14 | 87.57 | 85.25 | 82.46 |
| S5 new (Smith et al., 2023) | 62.15 | **89.31** | **91.40** | 88.00 | **95.33** | **98.58** | **87.46** |
| **Liquid-S4**-KB (ours) | 62.55 (0.10) | 88.97 (0.02) | 91.10 (0.05) | 89.37 (0.22) | 94.50 (0.08) | 96.10 | 87.09 |
| **Liquid-S4**-PB (ours) | **62.75** (0.20) | 89.02 (0.04) | 91.20 (0.01) | **89.50** (0.40) | 94.80 (0.20) | 96.66 | 87.32 |
| | p = 5 | p=6 | p=2 | p=3 | p=2 | p=2 | |

**Computational Complexity of the Liquid-S4 Kernel.** The computational complexity of the S4-Legs Convolutional kernel solved via the Cauchy Kernel is $\tilde{\mathcal{O}}(N + L)$, where N is the state-size, and L is the sequence length [Gu et al. (2022a), Theorem 3]. Liquid-S4 both in KB and PB modes can be computed in $\tilde{\mathcal{O}}(N + L + p_{max}\tilde{L})$. The added time complexity in practice is tractable. This is because we usually select the liquid orders, $p$, to be less than 10 (typically $p_{max} = 3$, and $\tilde{L}$ which is the number of terms we use to compute the input correlation vector, $u_{correlation}$, is typically two orders of magnitude smaller than the sequence length.

## 4 EXPERIMENTS WITH LIQUID-S4

In this section, we present an extensive evaluation of Liquid-S4 on sequence modeling tasks with very long-term dependencies and compare its performance to a large series of baselines ranging from advanced Transformers and Convolutional networks to many variants of state-space models. In the following, we first outline the baseline models we compare against. We then list the datasets we evaluated these models on and finally present results and discussions.

**Baselines.** We consider a broad range of advanced models to compare Liquid-S4 with. These baselines include transformer variants such as vanilla and Sparse Transformers, a Transformer model with local attention, Longformer, Linformer, Reformer, Sinkhorn, BigBird, Linear Transformer, and Performer. We also include architectures such as FNets, Nyströmformer, Luna-256, H-Transformer-1D, and Circular Diluted Convolutional neural networks (CDIL). We then include a full series of state-space models and their variants such as diagonal SSMs (DSS), S4, S4-legS, S4-FouT, S4-LegS/FouT, S4D-LegS, S4D-Inv, S4D-Lin and the Simplified structured state-space models (S5).

**Datasets.** We first evaluate Liquid-S4's performance on the well-studied **Long Range Arena (LRA)** benchmark (Tay et al., 2020b), where Liquid-S4 outperforms other S4 and S4D variants in every task with an average accuracy of **87.32%**. LRA dataset includes six tasks with sequence lengths ranging from 1k to 16k. Concurrent work We then report Liquid-S4's performance compared to other S4, and S4D variants, as well as other models, on the **BIDMC Vital Signs** dataset (Pimentel et al.,

2016; Goldberger et al., 2000). BIDMC uses bio-marker signals of length 4000 to predict Heart rate (HR), respiratory rate (RR), and blood oxygen saturation (SpO2). We also experiment with the **sCIFAR dataset** that consists of the classification of flattened images in the form of 1024-long sequences into ten classes.

Finally, we perform **Raw Speech Command (SC) recognition with full 35 labels** as conducted very recently in the updated S4 article (Gu et al., 2022a). It is essential to denote that there is a modified speech command dataset that restricted the dataset to only ten output classes and is used in a couple of works (see for example Kidger et al. (2020); Gu et al. (2021); Romero et al. (2021b;a)). Aligned with the updated results reported in Gu et al. (2022a) and Gu et al. (2022b), we choose not to break down this dataset and use the full-sized benchmark. SC dataset contains sequences of length 16k to be classified into 35 commands. Gu et al. (2022a) introduced a new test case setting to assess the performance

Table 2: Performance on BIDMC Vital Signs dataset. Numbers indicate RMSE on the test set. Models denoted by * is reported from (Gu et al., 2022b). The rest of the models' performance results are reported from the cited paper.

| | **BIDMC** | | |
|---|---|---|---|
| Model | HR | RR | SPO2 |
| UnICORNN (Rusch & Mishra, 2021b) | 1.39 | 1.06 | 0.869 |
| coRNN (Rusch & Mishra, 2021a) | 1.81 | 1.45 | - |
| CKConv* | 2.05 | 1.214 | 1.051 |
| NRDE (Morrill et al., 2021) | 2.97 | 1.49 | 1.29 |
| LSTM (Rusch & Mishra, 2021b) | 10.7 | 2.28 | - |
| Transformer* | 12.2 | 2.61 | 3.02 |
| XGBoost (Tan et al., 2021) | 4.72 | 1.67 | 1.52 |
| Random Forest (Tan et al., 2021) | 5.69 | 1.85 | 1.74 |
| Ridge Regress. (Tan et al., 2021) | 17.3 | 3.86 | 4.16 |
| S4-LegS* (Gu et al., 2022b) | 0.332 (0.013) | 0.247 (0.062) | 0.090 (0.006) |
| S4-FouT* (Gu et al., 2022b) | 0.339 (0.020) | 0.301 (0.030) | 0.068 (0.003) |
| S4D-LegS* (Gu et al., 2022b) | 0.367 (0.001) | 0.248 (0.036) | 0.102 (0.001) |
| S4-(LegS/FouT)* (Gu et al., 2022b) | 0.344 (0.032) | 0.163 (0.008) | 0.080 (0.007) |
| S4D-Inv* (Gu et al., 2022b) | 0.373 (0.024) | 0.254 (0.022) | 0.110 (0.001) |
| S4D-Lin* (Gu et al., 2022b) | 0.379 (0.006) | 0.226 (0.008) | 0.114 (0.003) |
| **Liquid-S4**-KB (ours) | 0.310 (0.001) | 0.162 (0.001) | 0.068 (0.002) |
| **Liquid-S4**-PB (ours) | **0.303** (0.002) | **0.158** (0.001) | **0.066** (0.002) |
| | p=3 | p=2 | p=4 |

of models (trained on 16kHz sequences) on sequences of length 8kHz. S4 and S4D perform exceptionally well in this zero-shot test scenario.

## 4.1 Results on Long Range Arena

Table 6 depicts a comprehensive list of baselines benchmarked against each other on six long-range sequence modeling tasks in LRA. We observe that Liquid-S4 instances (all use the PB kernel with a scaled Legendre (LegS) configuration) with a small liquid order, $p$, ranging from 2 to 6, consistently outperform all baselines in all six tasks, establishing the new SOTA on LRA with an average

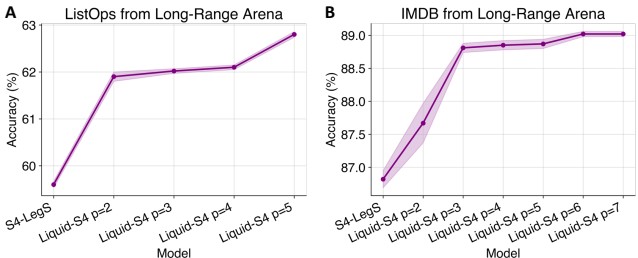

Figure 1: Performance vs Liquid Order in Liquid-S4 for A) ListOps, and B) IMDB datasets. More in Appendix. (n=3)

performance of **87.32%**. In particular, on ListOps, Liquid-S4 improves S4-LegS performance by more than 3%, on character-level IMDB by 2.2%, and on 1-D pixel-level classification (CIFAR) by 0.65%, while establishing the-state-of the-art on the hardest LRA task by gaining **96.66%** accuracy. Liquid-S4 performs on par with improved S4 and S4D instances on both AAN and Pathfinder tasks. The performance of SSM models is generally well-beyond what advanced Transformers, RNNs, and Convolutional networks achieve on LRA tasks, with the Liquid-S4 variants standing on top.

**The impact of increasing Liquid Order $p$.** Figure 1 illustrates how increasing the liquid order, $p$, can improve performance on ListOps and IMDB tasks from LRA (More results in Appendix).

## 4.2 Results on BIDMC Vital Signs

Table 2 demonstrates the performance of a variety of classical and advanced baseline models on the BIDMC dataset for all three heart rate (HR), respiratory rate (RR), and blood oxygen saturation (SpO2) level prediction tasks. We observe that Liquid-s4 with a PB kernel of order $p = 3$, $p = 2$, and $p = 4$, perform better than all S4 and S4D variants. It is worth denoting that Liquid-S4 is built by the same parametrization as S4-LegS (which is the official S4 model reported in the updated S4 report (Gu et al., 2022a)). In RR, Liquid-S4 outperforms S4-LegS by a significant margin of 36%. On SpO2, Liquid-S4 performs 26.67% better than S4-Legs. On HR, Liquid-S4 outperforms S4-Legs by 8.7% improvement in performance.

### 4.3 Results on Image Classification

Similar to the previous tasks, a Liquid-S4 network with PB kernel of order $p = 3$ outperforms all variants of S4 and S4D while being significantly better than Transformer and RNN baselines as summarized in Table 3.

### 4.4 Results on Speech Commands

Table 4 demonstrates that Liquid-S4 with $p = 2$ achieves the best performance amongst all benchmarks on the 16KHz testbed. Liquid-S4 also performs competitively on the half-frequency zero-shot experiment, while it does not realize the best performance. Although the task is solved to a great degree, the reason could be that liquid kernel accounts for covariance terms. This might influence the learned representations in a way that hurts performance by a small margin in this zero-shot experiment. The hyperparameters are given in Appendix.

Table 3: Performance on sCIFAR. Numbers indicate Accuracy (standard deviation). The baseline models are from Table 9 of Gu et al. (2022b).

| Model | Accuracy |
|---|---|
| Transformer (Trinh et al., 2018) | 62.2 |
| FlexConv (Romero et al., 2021a) | 80.82 |
| TrellisNet (Bai et al., 2018) | 73.42 |
| LSTM | 63.01 |
| r-LSTM (Trinh et al., 2018) | 72.2 |
| UR-GRU (Gu et al., 2020b) | 74.4 |
| HiPPO-RNN (Gu et al., 2020a) | 61.1 |
| LipschitzRNN (Erichson et al., 2021) | 64.2 |
| S4-LegS (Gu et al., 2022b) | 91.80 (0.43) |
| S4-FouT (Gu et al., 2022b) | 91.22 (0.25) |
| S4-(LegS/FouT) (Gu et al., 2022b) | 91.58 (0.17) |
| S4D-LegS (Gu et al., 2022b) | 89.92 (1.69) |
| S4D-Inv (Gu et al., 2022b) | 90.69 (0.06) |
| S4D-Lin (Gu et al., 2022b) | 90.42 (0.03) |
| S5 Smith et al. (2022) | 89.66 |
| Liquid-S4-KB (ours) | 91.86 (0.08) |
| Liquid-S4-PB (ours) | **92.02** (0.14) |
| | p=3 |

It is essential to denote that there is a modified speech command dataset that restricts the dataset to only ten output classes, namely SC10, and is used in a couple of works (see for example (Kidger et al., 2020; Gu et al., 2021; Romero et al., 2021b;a)). Aligned with the updated results reported in (Gu et al., 2022a) and (Gu et al., 2022b), we choose not to break down this dataset and report the full-sized benchmark in the main paper. Nevertheless, we conducted an experiment with SC10 and showed that even on the reduced dataset, with the same hyperparameters, we solved the task with a SOTA accuracy of 98.51%. The results are presented in Table 7.

## 5 CONCLUSIONS

We showed that the performance of structured state-space models could be considerably improved if they are formulated by a linear liquid time-constant kernel, namely Liquid-S4. Liquid-S4 kernels are obtainable with minimal effort, with their kernel computing the similarities between time-lags of the input signals in addition to the main S4 diagonal plus low-rank parametrization. Liquid-S4 kernels with smaller parameter counts achieve SOTA performance on all six tasks of the Long-range arena dataset, on BIDMC heart rate, respiratory rate, and blood oxygen saturation, on sequential 1-D pixel-level image classification, and on Speech command recognition. As a final note, our experimental evaluations suggest that for challenging multivariate time series and modeling complex signals with long-range dependencies, SSM variants such as Liquid-S4 dominate other baselines, while for image and text data, a combination of SSMs and attention might enhance model quality.

Table 4: Performance on Raw Speech Command dataset with **Full 35 Labels**. Numbers indicate Accuracy on test set. The baseline models are reported from Table 11 of (Gu et al., 2022b).

| Model | Parameters | 16000Hz | 8000Hz |
|---|---|---|---|
| InceptionNet (Nonaka & Seita, 2021) | 481K | 61.24 (0.69) | 05.18 (0.07) |
| ResNet-18 | 216K | 77.86 (0.24) | 08.74 (0.57) |
| XResNet-50 | 904K | 83.01 (0.48) | 07.72 (0.39) |
| ConvNet | 26.2M | 95.51 (0.18) | 07.26 (0.79) |
| S4-LegS (Gu et al., 2022b) | 307K | 96.08 (0.15) | 91.32 (0.17) |
| S4-FouT (Gu et al., 2022b) | 307K | 95.27 (0.20) | 91.59 (0.23) |
| S4-(LegS/FouT) (Gu et al., 2022b) | 307K | 95.32 (0.10) | 90.72 (0.68) |
| S4D-LegS (Gu et al., 2022b) | 306K | 95.83 (0.14) | 91.08 (0.16) |
| S4D-Inv (Gu et al., 2022b) | 306K | 96.18 (0.27) | **91.80** (0.24) |
| S4D-Lin (Gu et al., 2022b) | 306K | 96.25 (0.03) | 91.58 (0.33) |
| Liquid-S4-KB (ours) | 224K | 96.52 (0.04) | 91.30 (0.19) |
| Liquid-S4-PB (ours) | 224K | **96.78** (0.05) | 90.00 (0.25) |
| | | p=2 | p=2 |

## ACKNOWLEDGMENTS

This research was supported in part by the AI2050 program at Schmidt Futures (Grant G-22-63172) and the United States Air Force Artificial Intelligence Accelerator under Cooperative Agreement Number FA8750-19-2-1000. The views and conclusions contained in this document are those of the authors and should not be interpreted as representing the official policies, either expressed or implied, of the United States Air Force or the U.S. Government. The U.S. Government is authorized to reproduce and distribute reprints for Government purposes, notwithstanding any copyright notation herein. We are very grateful.

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

# A    EXAMPLE LIQUID-S4 KERNEL

$$\overline{\mathbf{K}}_{\text{liquid}} * u_{\text{correlations}} = \left[ \overline{\mathbf{CA}}^{(k-1)}\overline{\mathbf{B}}^2, \ldots, \overline{\mathbf{CB}}^2, \ldots, \overline{\mathbf{CA}}^{(k-2)}\overline{\mathbf{B}}^3, \ldots, \overline{\mathbf{CB}}^3, \ldots, \overline{\mathbf{CA}}^{(k-3)}\overline{\mathbf{B}}^4, \ldots, \overline{\mathbf{CB}}^4 \right] * \quad (16)$$

$$\left[ u_0 u_1, \ldots, u_{k-1} u_k, \ldots, u_0 u_1 u_2, \ldots, u_{k-2} u_{k-1} u_k, \ldots, u_0 u_1 u_2 u_3, \ldots, u_{k-3} u_{k-2} u_{k-1} u_k \right]^T$$

Here, $u_{\text{correlations}}$ is a vector of length $\binom{k+1}{2} + \binom{k+1}{3} + \binom{k+1}{4}$, and the kernel $\overline{\mathbf{K}}_{\text{liquid}} \in \mathbb{R}^{\binom{k+1}{2}+\binom{k+1}{3}+\binom{k+1}{4}}$.

# B    PROOF OF PROPOSITION 1

**Proposition.**    The Liquid-S4 kernel for each order $p \in \mathcal{P}$, $\overline{\mathbf{K}}_{\text{liquid}}$, can be computed by the anti-diagonal transformation (flip operation) of the product of the S4 convolution kernel, $\overline{\mathbf{K}} = \left(\overline{\mathbf{CB}}, \overline{\mathbf{CAB}}, \ldots, \overline{\mathbf{CA}}^{L-1}\overline{\mathbf{B}}\right)$, and a vector $\overline{\boldsymbol{B}}^{p-1} \in \mathbb{R}^N$.

*Proof.* This can be shown by unrolling the S4 convolution kernel and multiplying its components with $\overline{\boldsymbol{B}}^{p-1}$, performing an anti-diagonal transformation to obtain the corresponding liquid S4 kernel:

$$\overline{\mathbf{K}} = \left(\overline{\mathbf{CB}}, \overline{\mathbf{CAB}}, \overline{\mathbf{CA}}^2\overline{\mathbf{B}}, \ldots, \overline{\mathbf{CA}}^{L-1}\overline{\mathbf{B}}\right)$$

For $p = 2$ (correlations of order 2), S4 kernel should be multiplied by $\overline{\boldsymbol{B}}$. The resulting kernel would be:

$$\left(\overline{\mathbf{CB}}^2, \overline{\mathbf{CAB}}^2, \overline{\mathbf{CA}}^2\overline{\mathbf{B}}^2, \ldots, \overline{\mathbf{CA}}^{L-1}\overline{\mathbf{B}}^2\right)$$

We obtain the liquid kernel by flipping the above kernel to be convolved with the 2-term correlation terms (p=2):

$$\overline{\mathbf{K}}_{\text{liquid=2}} = \left(\overline{\mathbf{CA}}^{L-1}\overline{\mathbf{B}}^2, \ldots, \overline{\mathbf{CA}}^2\overline{\mathbf{B}}^2, \overline{\mathbf{CAB}}^2, \overline{\mathbf{CB}}^2\right)$$

Similarly, we can obtain liquid kernels for higher liquid orders and obtain the statement of the proposition.    □

# C    HYPERPARAMETERS

**Learning Rate.**    Liquid-S4 generally requires a smaller learning rate compared to S4 and S4D blocks.

**Setting $\Delta t_{max}$ and $\Delta t_{min}$**    We set $\Delta t_{max}$ for all experiments to 0.2, while the $\Delta t_{min}$ was set based on the recommendations provided in (Gu et al., 2022c) to be proportional to $\propto \frac{1}{\text{seq length}}$.

**Causal Modeling vs. Bidirectional Modeling**    Liquid-S4 works better when it is used as a causal model, i.e., with no bidirectional configuration.

$d_s tate$    We observed that Liquid-S4 PB kernel performs best with smaller individual state sizes $d_s tate$. For instance, we achieve SOTA results in ListOps, IMDB, and Speech Commands by a state size set to 7, significantly reducing the number of required parameters to solve these tasks.

**Choice of Liquid-S4 Kernel**    In all experiments, we choose our simplified PB kernel over the KB kernel due to the computational costs and performance. We recommend the use of PB kernel.

**Choice of parameter $p$ in liquid kernel.**    In all experiments, start off by setting $p$ or the liquidity order to 2. This means that the liquid kernel is going to be computed only for correlation terms of order 2. In principle, we observe that higher $p$ values consistently enhance the representation learning capacity of Liquid-S4 modules, as we showed in all experiments. We recommend $p = 3$ as a norm to perform experiments with Liquid-S4.

Table 6: Performance on Long Range Arena Tasks. Numbers for Liquid-S4 kernels indicate **test accuracy** (standard deviation). The rest of the models' performance results are reported from the cited paper. Liquid-S4 is used with its PB kernel.

| Model (input length) | ListOps 2048 | IMDB 2048 | AAN 4000 | CIFAR 1024 | Pathfinder 1024 | Path-X 16384 | Avg. |
|---|---|---|---|---|---|---|---|
| S4-LegS (Gu et al., 2022b) | 59.60 (0.07) | 86.82 (0.13) | 90.90 (0.15) | 88.65 (0.23) | 94.20 (0.25) | 96.35 | 86.09 |
| S4D-LegS (Gu et al., 2022b) | 60.47 (0.34) | 86.18 (0.43) | 89.46 (0.14) | 88.19 (0.26) | 93.06 (1.24) | 91.95 | 84.89 |
| S4D-Inv (Gu et al., 2022b) | 60.18 (0.35) | 87.34 (0.20) | 91.09 (0.01) | 87.83 (0.37) | 93.78 (0.25) | 92.80 | 85.50 |
| **Liquid-S4**-KB (ours) | 62.30 (0.10) | 88.80 (0.05) | 90.95 (0.10) | 89.40 (0.15) | 94.60 (0.10) | 95.98 | 87.01 |
| **Liquid-S4**-PB (ours) | **62.60** (0.20) | **88.90** (0.10) | **91.15** (0.09) | **89.45** (0.33) | **94.90** (0.25) | **96.36** | **87.21** |
| | p = 5 | p=6 | p=4 | p=3 | p=2 | p=2 | |

The kernel computation pipeline uses the PyKeops package (Charlier et al., 2021) for large tensor computations without memory overflow.

All reported results are validation accuracy (similar to Gu et al. (2022a)) performed with 2 to 3 different random seeds, except for the BIDMC dataset, which reports accuracy on the test set.

Table 5: Hyperparameters for obtaining best performing models. BN= Batch normalization, LN = Layer normalization, WD= Weight decay.

| | Depth | Features $H$ | State Size | Norm | Pre-norm | Dropout | LR | Batch Size | Epochs | WD |
|---|---|---|---|---|---|---|---|---|---|---|
| **ListOps** | 9 | 128 | 7 | BN | True | 0.01 | 0.002 | 12 | 30 | 0.03 |
| **Text (IMDB)** | 4 | 128 | 7 | BN | True | 0.1 | 0.003 | 8 | 50 | 0.01 |
| **Retrieval (AAN)** | 6 | 256 | 64 | BN | False | 0.2 | 0.005 | 16 | 20 | 0.05 |
| **Image (CIFAR)** | 6 | 512 | 512 | LN | False | 0.1 | 0.01 | 16 | 200 | 0.03 |
| **Pathfinder** | 6 | 256 | 64 | BN | True | 0.0 | 0.0004 | 4 | 200 | 0.03 |
| **Path-X** | 6 | 320 | 64 | BN | True | 0.0 | 0.001 | 8 | 60 | 0.05 |
| **Speech Commands** | 6 | 128 | 7 | BN | True | 0.0 | 0.008 | 10 | 50 | 0.05 |
| **BICMD (HR)** | 6 | 128 | 256 | LN | True | 0.0 | 0.005 | 32 | 500 | 0.01 |
| **BICMD (RR)** | 6 | 128 | 256 | LN | True | 0.0 | 0.01 | 32 | 500 | 0.01 |
| **BICMD (SpO2)** | 6 | 128 | 256 | LN | True | 0.0 | 0.01 | 32 | 500 | 0.01 |
| **sCIFAR** | 6 | 512 | 512 | LN | False | 0.1 | 0.01 | 50 | 200 | 0.03 |

Table 7: Performance on Raw Speech Command dataset with the reduced ten classes (SC10) dataset. Numbers indicate validation accuracy. The accuracy of baseline models is reported from Table 5 of (Gu et al., 2022a). x stands for infeasible computation on a single GPU or not applicable as stated in Table 10 of (Gu et al., 2022a). The hyperparameters for Liquid-S4 are the same as the ones reported for Speech Commands Full Dataset reported in Table 5.

| | SC10 | |
|---|---|---|
| Model | 16kHz | 8kHz |
| Transformer | x | x |
| Performer | 30.77 | 30.68 |
| ODE-RNN | x | x |
| NRDE | 16.49 | 15.12 |
| ExpRNN | 11.6 | 10.8 |
| LipschitzRNN | x | x |
| CKConv | 71.66 | 65.96 |
| WaveGAN-D | 96.25 | x |
| LSSL (Gu et al., 2021) | x | x |
| S4-LegS (Gu et al., 2022a) | 98.32 | **96.30** |
| Liquid-S4 (ours) | **98.51** | 95.9 |
| | p=2 | p=2 |

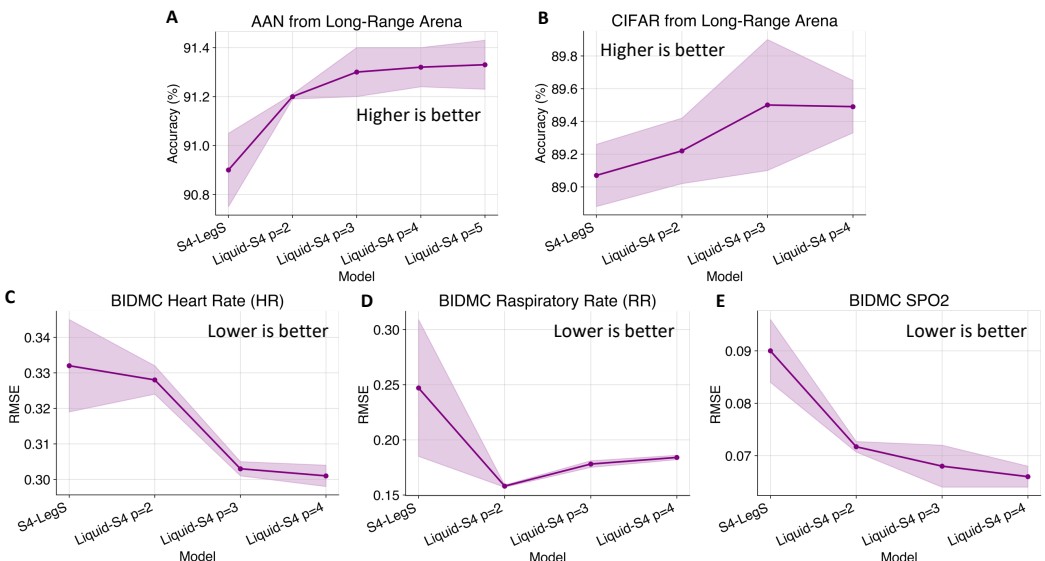

Figure 2: Performance vs Liquid Order in Liquid-S4. (n=3)

## D   ON THE DISCRETIZATION OF THE LIQUID KERNEL.

How do we perform the discretization of $A + Bu(t)$. The dynamical system presented in Eq. 10, is a continuous-time (CT) bilinear state-space model (SSM). Ideally, we want that the discretization of a CT bilinear SSM to 1) satisfy the first-order form of the model, and 2) preserve the bilinear model structure. This is challenging and only possible via a limited number of methods:

1) The most straightforward approach is to use Forward Euler with first-order error: $\dot{x} = \frac{x_{k+1}-x_k}{\delta t} + \mathcal{O}(\delta t)$. Now by plugging this in Eq. 10, we get $\overline{A} = I + \delta t A, \overline{B} = \delta t B$. Such discretization satisfies the conditions above. For this discretization to stay stable, for $s = \sigma \pm i\omega$ an eigenvalue of the continuous transition matrix A, and $\lambda = 1 + s\delta t$, an eigenvalue of the discrete model, $\mathcal{R}e(s) \leq 0$ or $|\lambda| = |1 + s\delta t| \leq 1$, thus $(1 + \sigma\delta t)^2 + \omega^2\delta t^2 \leq 1$. This condition implies that selecting a small enough $\delta t$ ensures the system's stability, but for cases where $\delta t$ is large, the system might go unstable.

One can show that based on the properties of the transition matrices A and B, and the range of selected $\delta t$ a bilinear transformation of discrete matrices $\overline{A}$ and $\overline{B}$, would be very close to that of our Forward Euler discretization. This means that:

$$|\overline{A}_{\text{Forward Euler}} - \overline{A}_{\text{Approx bilinear}}| < \gamma \quad 0 < \gamma < \frac{\delta t}{2} \tag{17}$$

2) Adams-Bashforth Method: The second-order Adams-Bashforth will apply the transformation $x_{k+1} = x_k + \frac{3\delta t}{2}f(k) - \frac{\delta t}{2}f(k-1) + \mathcal{O}(\delta t^2)$, where f(k) is the right-hand-side of Eq. 8 at time $t = k\delta t$. This method also satisfies the two conditions we required (Phan et al., 2012).

One must denote that computing a bilinear transform (https://en.wikipedia.org/wiki/Bilinear_transform) of a continuous-time bilinear SSM while preserving the first-order structure of the model is an open problem. Ideally, we can apply this transformation on A+Bu(t). However, it is challenging to preserve the first-order form of the equation while keeping the bilinear (liquid) structure of the model described in Eq. 10.

In our case, we use the bilinear transform form of A, B, and C presented in (Eq. 3) for the discrete weights of the system of (Eq. 11), as this approximation is close to that of the Forward Euler. This implies that the continuous system in (Eq. 10) could be transformed directly to Eq. 11 by a forward Euler transformation. Furthermore, due to the range of $\delta t$ and properties of A and B, the bilinear transformed matrices presented in Eq. 3, would be close to the direct forward Euler system.

# E   ON THE AUTOREGRESSIVE MODE OF PB KERNEL

In autoregressive (AR) mode, with PB (or any other conditioned kernel), we obtain $\overline{A}$, $\overline{B}$, and $\overline{C}$ no matter what the conditions are.

More specifically, in the autoregressive mode of PB we can use Eq. 13. In Eq. 13, for computing the black parts we can reuse the AR mode of the plain S4 model and only have computed the new (violet) parts. As the violet parts consist of p multiplications of input terms (and the corresponding matrices) computing the AR mode is feasible. This adds a complexity of O(p) to the inference in the AR mode of PB, but because p is much smaller than L (past sequence length), it can significantly speed up the inference time compared to the convolution counterpart.

Moreover, one of the properties of LTCs that was never studied and introduced before this work is their ability to account for the pairwise correlation of inputs which became apparent once we unrolled the system's dynamics in this work. We believe that the pairwise correlation of inputs is a property that the PB kernel also possesses. Whether the kernel loses the expressivity and robustness attributes of LTCs, we have to investigate in future work.

# F   WHY DOES PB KERNEL OUTPERFORM KB KERNEL?

One possible reason why PB outperforms KB could be the fact that we limit the correlation terms with the truncation with order p. This limitation arises from how S4 blocks are constructed as a stack of many 1D blocks which does not computationally allow us to exploit the benefit of higher-order correlation terms due to the high computational complexity. This limitation might also reduce the expressivity of the KB kernel, but not PB, as they do not have a dependency on A for the correlation terms.

A potential solution would be a Liquid-S5 instance, where we could directly use a parallel scan introduced in the concurrent work S5 (Smith et al., 2023), over the linear LTC system (which is a time-varying SSM). This is possible because we could precompute the state transitions at each time step. This way we would not need to truncate the kernel and obtain all correlation terms for free. This is an exciting extension to Liquid-S4 which we are exploring in future work

