# OpenReview forum: "Liquid Structural State-Space Models"
_ICLR.cc/2023/Conference — ICLR 2023 poster_

### Official Review · Reviewer_WFdy · 2022-10-22

**Confidence:** 5
**Correctness:** 3
**Technical Novelty And Significance:** 2
**Empirical Novelty And Significance:** 2
**Recommendation:** 5

**Clarity, Quality, Novelty And Reproducibility:**

The work is not especially well presented.  The prose is littered with typographical mistakes and sentences that don’t read especially well.  This makes the paper feel very rushed.  The prose is also very long in places.  The paper would benefit from cutting a lot of the “fat” from the main text (see (a) above).

The work is novel in the sense that no one has combined LTC models with S4.  Deriving the efficient form for the liquid kernel is nice.  However, the paper does have somewhat of an incremental feel to it.

I believe that the work would be fully reproducible.


**Strength And Weaknesses:**


## Strengths
The method appears to perform very strongly on a good range of standard benchmarks.  The method also combines two existing methods nicely.  The (short) derivation of how the truncated liquid kernel can be computed, and its parallels to the S4 kernel are also nice.  Highlighting how (effectively) time-varying kernels can be used in S4 is nice.

## Weaknesses
My main criticisms with the paper are that it is well below the threshold in terms of presentation, some questions on the core of the model, and that the contribution feels a little incremental.

(a) There are numerous typos, typographical errors, inconsistencies etc that make the paper feel incredibly rushed.  I have tried to outline some of these below.

The paper is also very wordy for what is quite a simple idea.  It feels like the authors have engineered paragraphs to cram dozens of citations in to make the work feel more thoroughly referenced; when really the work is just LTC models + S4.  For instance, listing the baselines and their citations in the paragraph “Baselines” is totally unnecessary (since they are also in Table 1).  The same criticism goes for much of Section 2.

I think this paper could very easily be reduced to say six pages, and that a shorter manuscript would dramatically improve its impact.

I also wished the authors separated S4 into its own background section.  Although the core concepts predate S4, the math in eg. (4) clearly is drawn from S4.  In that sense, I think “Background” should actually be called “S4”.  Similarly, the bullet points in “How to compute LS4 Kernel Efficiently” should also be in this background section.

Really, I feel like your sections should be:  “S4”, “LTCMs”, and then the methods section is just “S4+LTCMs”.  This will dramatically simplify the exposition for the reader, and make it clear what is a novel contribution in **this** paper.

(b)  Can the authors clarify some things in the relation to LCTMs for me:  You actually use the "PB" model variant in all of your experiments, which means that you aren't actually using the discrete-time LTC model in (9).  This should be highlighted in the table, where your method is labelled as "Liquid-S4", as opposed to "Liquid-S4-PB".  Is there a reason why you don't evaluate "KB"?  Secondly, I am unconvinced that the discretisation from (8) to (9) is correct.  The bilinear (or ZOH) transform is non-linear, and so discretising (8) is not as simple as discretising $A$ and $B$ separately.  I think the correct discretisation is $\overline{A} \cdot \overline{(Bu_k)}$.  This scheme would make computing the recurrence in (11) more difficult.  I am willing to entertain that I have mis-evaluated this, but this means that the model for which results are presented is actually quite a drastic deviation from the initially motivated LS4 model.  This is okay, but does mean that much of the prose up until (15) is a touch misleading.  I invite the authors to comment on this.

I also query if the learned PB model can be iterated in a recurrent mode for autoregressive generation?  If this is not the case, then a major plus-point of the original S4 work has been lost.

(c)  My final major criticism is that the work feels a little incremental.  If we believe the extra cross-terms in (linear) LTC models are useful, as prior work has demonstrated, then there is no reason to think they won’t improve S4 models.  Adding more flexibility into a model should improve performance.  The only “novel” work therefore is showing that these cross-terms can be cast in the same terms as the S4 update, which is nice, but is also not totally surprising given the nature of the cross-terms.

The ablation presented in Figure 1 is useful.  I would like to see the ablation for all the datasets:  did increasing p degrade performance?  Or did performance plateau?  What happens if we consider p=100?

Do any of the theoretical results from S4/DSS/S4D carry across to LS4?  If not, why not?  Is tying the $B$ matrices in the transition and input important?

Side note on Figure 1:  why was p=4 not evaluated?

Are you also able to clarify whether the figures in the table are the validation or test scores?



## Minor Weaknesses / Typographical Comments.
(i)  Throughout:  Inconsistent use of \eqref{}, \ref{}, Equ. \ref{} etc.

(ii)  Throughout:  The math is weirdly spaced, cf. (1) or the second line on page two.

(iii)  Throughout:  Only proper nouns should be capitalized.  (i.e. “State” -> “state”)

(iv)  Throughout:  $x$ (and other terms in select places) should probably be boldfaced as they are vectors.

(v)  Reference:  “KALMAN” -> “Kalman”

(vi)  Pg 2:  “parameter” -> “parameters”.

(vii)  Pg 3:  The logic in attributing the performance of S4 isn’t quite correct.  Numerical performance is derived from HiPPO.  Memory and runtime performance is derived from the NPLR representation.

(viii)  Eqn. (8):  The element-wise addition is very non-standard.  I would include multiplication with a row vector of ones to signify this in math, as opposed to in the text.

(ix)  Alg 1:  The subscript “liquid” was in \mathrm elsewhere.

(x)  Eqn. (11):  The statement of this is far too informal.  It obfuscates the message.  Please write out the term fully in math.

(xi)  Throughout:  “Structural” -> “structured”  (in the context of S4).

(xii)  Throughout:  Footnotes should be avoided at all costs.

(xiii)  Throughout (cf. footnote 2):  \citep and \citet are used incorrectly in places.

(xiv)  RAW and FULL do not need to be capitalised.

(xv)  Citations are inconsistent in their use of initial, full first name, middle initial etc.  URLs should also be removed.

(xvi)  Appendix B:  it is nice to re-state the proposition prior to proving it.

(xvii)  Figure 1:  text-wrapped figures should be avoided.

(xviii):  You used an extra layer (9), compared to S4d's eight, in the ListOps experiment.  Why?  Is it possible to re-run LS4 with eight layers?


**Summary Of The Paper:**

This paper combines liquid time constant models with the recently published S4 model.  This extends S4 to have input-dependent dynamics.  This can be cast back into the original convolutional framework, and the efficient Cauchy kernel can be re-used for computing the additional terms.  The method achieves good results on the standard benchmarks.


**Summary Of The Review:**

The results in this paper speak for themselves.  However, the paper offers limited additional intuition or insight beyond “existing LTC ideas can improve S4”.  I also have reservations that the LTC ideas are carried across correctly as is currently presented.  This, coupled with the below-par presentation mean I am scoring this paper as a reject.  I am willing to upgrade my review if that authors can allay my concerns.

I also note that this is one of as many as six S4 extensions submitted to ICLR.  I would expect to see these at least commented on (although certainly not directly compared to) in a revised draft.

---

> ### Author Response · Authors · 2022-11-13
> **Response to Reviewer WFdy - Part 1**
>
> ## Part 1
> Thank you for the evaluation of our work in great detail and for providing constructive suggestions. We are very happy that the reviewer is willing to upgrade their score should their main concerns are addressed. We have prepared a response letter and updated our manuscript in parts accordingly to extensively address your concerns to encourage a positive evaluation of our work similar to the other 3 reviewers:
>
> The reviewer raises three main points of weakness:
>
> (a) Presentation matters
>
> (b) some questions on the core of the model
>
> (c) the contribution feels a little incremental
>
> --------
>
> (a) **Presentation matters:**
>
> Thank you for pointing out the typographical errors and suggesting a better structure for our work.
> The citation format is that of ICLR, and we simply tried to be inclusive of the related works! And thus the overload of references in Section 2, which we believe is unavoidable! But, we agree and simplified the “baselines” paragraph to enhance readability as suggested.
>
> **On the reorganization of the background section:** Thank you for this suggestion which indeed helps the paper be better organized. We have now addressed this in the revised version. Section 3.1 is now background: S4, and 3.2: Liquid-S4 model.
> The reviewer says: “I think this paper could very easily be reduced to say six pages, and that a shorter manuscript would dramatically improve its impact.” We respectfully disagree. For a reader that is not a subject matter expert, we must try our best to provide a standalone piece of work to enable them to follow our discussions. We believe that we did a successful job on that front as mentioned by all three other reviewers.
>
> **Typographical and Minor weaknesses comments** We would like to thank you very much for such a detailed read of our work and for pointing out the issues. We worked on the clarity issues and tried our best to incorporate all typographical errors in the current revised manuscript as advised by the reviewer. Indeed the suggestions improved the quality of our presentation.
>
> (b) **Questions about Liquid-S4:**
>
> **Why did you present the results with PB only and not KB?** As we stated on Page 16 of our submission: We choose our simplified PB kernel over the KB kernel due to the computational costs and performance. Liquid-S4’s PB kernel is faster and on average provides slightly better performance, although KB compared to PB results are not statistically much different (See Below) That is the reason why we chose not to dilute the information in tables with various variants and instead consistently provide results for the PB kernel. The fact that the Liquid-S4 is computed by PB kernel is also highlighted repeatedly in the manuscript and in the caption of Table 1. We will include these results in our revised version.
>
> **Long Range Arena** results are provided for the same p for both KB and PB kernels.
> |Model| Listops | IMDB | AAN | CIFAR | Pathfinder |Path-X | Avg. |
> |---|---|---|---|---|---|---|---|
> |Liquid-S4-PB| 62.75 | 89.02| 91.20 | 89.50 | 94.8 | 96.66 | 87.32 |
> |Liquid-S4-KB| 62.55 | 88.97 | 91.10 | 89.37 | 94.5 | 96.1 | 87.09 |
> |  | p=5 | p= 6 | p= 2 | p=3 | p= 2 |  p=2 |  |
>
> **BIDMC** results are provided for the same p for both KB and PB kernels.
> |Model| HR | RR | SPO2 |
> |---|---|---|---|
> |Liquid-S4-PB| 0.303 | 0.158 | 0.066 |
> |Liquid-S4-KB| 0.310 | 0.162 | 0.068 |
> |  | p=3 | p= 2 | p= 4 |
>
> **sCIFAR** results are provided for the same p for both KB and PB kernels.
> |Model| Accuracy|
> |---|---|
> |Liquid-S4-PB| 92.02 |
> |Liquid-S4-KB| 91.86 |
> |  | p= 3 |
>
> **Speech Commands** results are provided for the same p for both KB and PB kernels.
> | Model | Parameters | 16k Hz | 8k Hz |
> |---|---|---|---|
> |Liquid-S4-PB| 224k | 96.78 | 90.00 |
> |Liquid-S4-KB| 224k | 96.52 | 91.3 |
> |  | p= 2 | p=2 |

---

> > ### Author Response · Authors · 2022-11-13
> > **Response to Reviewer WFdy - Part 2**
> >
> > ## Part 2
> >
> > **On the discretization of the system in Eq. 8 to Eq. 9:** The dynamical system presented in Eq. 8, is a continuous-time (CT) bilinear state-space model (SSM). Ideally, we want that the discretization of a CT bilinear SSM to 1) satisfy the first-order form of the model, and 2) preserve the bilinear model structure. This is challenging and only possible via a limited number of methods:
> >
> > 1) The most straightforward approach is to use Forward Euler with first-order error:
> >
> > $\dot{x} = \frac{x_{k+1} - x_{k} }{\delta t} + \mathcal{O}(\delta t)$. Now by plugging this in Eq. 8, we get $\overline{A} = I + \delta t A$, $\overline{B} = \delta t B $. Such discretization satisfies the conditions above. For this discretization to stay stable, for $s=\sigma \pm i \omega$ an eigenvalue of the continuous transition matrix A, and $\lambda = 1 + s \delta t $, an eigenvalue of the discrete model, $\mathcal{R}e(s) \leq 0$ or $|\lambda| = |1 + s \delta t | \leq 1$, thus $(1 + \sigma \delta t)^2 + \omega^2 \delta t^2 \leq 1$. This condition implies that selecting a small enough $\delta t$ ensures the system's stability, but for cases where $\delta t$ is large, the system might go unstable.
> >
> > We choose this straightforward Forward Euler discretization. We can show that based on the properties of the transition matrices A and B, and the range of selected $\delta t$ a separate bilinear transformation of discrete matrices $\overline{A}$ and $\overline{B}$, would be very close to that of our Forward Euler discretization. This means:
> >
> > $ |  \overline{A} _{\text{Forward Euler}} - \overline{A} _{\text{Approx bilinear}} | < \gamma ~~~~ 0 < \gamma < \frac{\delta t}{2}$
> >
> > 2) Adams-Bashforth Method: The second-order Adams-Bashforth will apply the transformation $x_{k+1} = x_k + \frac{3 \delta t}{2} f(k) - \frac{\delta t}{2} f(k-1) + \mathcal{O}(\delta t^2)$, where f(k) is the right-hand-side of Eq. 8 at time $t= k \delta t$. This method also satisfies the two conditions we required and could be used to discretize the CT bilinear SSM presented in Eq. 8.
> >
> > As the reviewer denoted as well, computing a bilinear transform of a continuous-time bilinear SSM while preserving the first-order structure of the model is difficult. one could apply the transformation directly on A+Bu(t). However, it is challenging to preserve the first-order form of the equation while keeping the bilinear (liquid) structure of the model described in Eq. 8. We will certainly add an appendix and import this discussion in the main text as well, in our revised version.
> >
> > **Can PB kernel be used in Autoregressive mode?** Liquid-S4 both in PB and KB can be used in the autoregressive (AR) mode. In these settings, we simply use the recurrent view of liquid-s4 illustrated by Eq. 8 and compute $\overline{A}$ and $\overline{B}$ by either forward Euler or the approximate forward-backward discretization. Similar to S4, with both discretization schemes, we can compute Liquid-S4 recurrence in O(N), where N is the state size.
> >
> > (c) **the work feels a little incremental:**
> >
> > Beyond major improvements in performance over S4, we showed **how** to integrate the two promising technologies Liquid networks and S4. We presented **efficient algorithms** for computing the hybrid kernels, and justified, **why** does a bilinear SSM should work better. Previous work on LTCs [1] studied the expressivity of this class of neural networks from a deep learning theory perspective (Trajectory length). In this work, we expanded a linearized LTC system from a dynamical system perspective and equipped it with the beautiful properties of structured SSMs.
> >
> > We are actually proud of the simplicity of our method, as indicated by all other reviewers, and how well and systematically it can help improve performance in structured SSMs.
> >
> > Finally, our work certainly inspires exciting future research in this promising direction to learning representations, by revealing how intriguing characteristics of dynamical systems can enable performant machine learning systems.

---

> > > ### Author Response · Authors · 2022-11-13
> > > **Response to Reviewer WFdy - Part 3**
> > >
> > > ## Part 3
> > > **The ablation presented in Figure 1 is useful. I would like to see the ablation for all the datasets: did increasing p degrade performance? Or did performance plateau? What happens if we consider p=100?**
> > >
> > > As requested by the reviewer, we have included new results for the ablation study with other datasets, including AAN, CIFAR, sCIFAR, and the BIMDC dataset. The results are presented in Figure 2 in our revised manuscript. We haven’t tried path-x with higher p simply because of compute limitations.
> > >
> > > The highest p we tried was with ListOps with p = 9, which plateaus.
> > >
> > > p= 100 would be the equivalent of computing the product of 100 time instances of the signal which would either explode or vanish. As we stated in the paper, with the current kernel implementations the maximum p that would give meaningful results is around 10. A potential solution to this might be extending the elegant parallelization scheme S5 provided to Liquid-S4 which would allow us to use all correlation terms. We will certainly look into this in future work.
> > >
> > > **Do any of the theoretical results from S4/DSS/S4D carry across to LS4? If not, why not?**
> > >
> > > Yes. Theorem 2 in the S4 paper directly applies to Liquid-S4 as well. Restating the theorem:
> > > Gu et al. 2022: “Theorem 2 (S4 Recurrence). Given any step size ∆, computing one step of the recurrence (3) can be done in O(N) operations where N is the state size.”
> > >
> > > For both Forward Euler discretization of Liquid-S4 and approximate bilinear discretization, following the same conjecture we can prove that Liquid-S4 Recurrence (Eq. 9) could be run in O(N). In the bilinear discretization form: $\overline{A}$ is obtained by the product of two DPLR matrices which was shown by Gu et al. 2022 that could be computed by O(N) operations.
> > > In the case of Forward Euler, it is even simpler as $\overline{A}$ is a solo forward discretization of a DPLR matrix and therefore, the matrix-vector multiplication in the recurrent view could be obtained in O(N).
> > > Theorem 3 on S4-Convolution kernel complexity of [Gu et. al. 2022]  is also applicable to Liquid-S4 as we stated in the paragraph: “Computational Complexity of the Liquid-S4 Kernel”
> > >
> > >
> > > **Is tying the B matrices in the transition and input important?**
> > >
> > > Choosing the same B as part of the autonomous system’s time constant 1/(A + Bu)  and input control creates a “coupling” effect in the LTCs’ original formulation (See Eq. 7), that appears to be an effective property for achieving better expressivity [1]. Inspired by the results presented in [1], we decided to preserve the structure of the LTC in the linearized version as well. Intuitively, selecting the same input transition matrix B simultaneously changes the characteristics (Time constant) of the LTC SSM and derives the input-dependent additive part of the ODE, therefore, establishing the coupling behavior.
> > >
> > > **Side note on Figure 1: why was p=4 not evaluated?**
> > >
> > > We simply did not evaluate p=4 and jumped to p=5 to explore higher orders. However, we performed additional experiments now to show the full range. Please see the updated fig 1 and the extended figures on other datasets in the appendix.
> > >
> > > **Are you also able to clarify whether the figures in the table are the validation or test scores?**
> > >
> > > All reported results on LRA are validation accuracy (similar to what is reported in Gu et al. (2022a)) performed with 2 to 3 different random seeds. We observed (similar to concurrent works such as S5 and other S4 variants) that the test accuracy and validation on LRA statistically reflect similar values on LRA tasks.
> > >
> > > On the BIDMC dataset, all reported results are test accuracy.
> > >
> > > On the Speech Command dataset, all reported results are test accuracy.
> > >
> > > **I also note that this is one of as many as six S4 extensions submitted to ICLR. I would expect to see these at least commented on (although certainly not directly compared to) in a revised draft.**
> > >
> > > Thank you for denoting this, and certainly. We will be mentioning these concurrent works in our revised version. Especially, when discussing these models we will also highlight how they can be potentially combined with our liquid-s4. For instance, a Liquid-S5 model could resolve the computational complexity of Liquid-S4 with higher order p values.
> > >
> > > **Minor weaknesses remaining**
> > >
> > > **(xviii): You used an extra layer (9), compared to S4d's eight, in the ListOps experiment. Why? Is it possible to re-run LS4 with eight layers?**
> > >
> > > Simply because we found that on ListOps and IMDB, Liquid-S4 kernels with slightly deeper structures but smaller state-size (s_state = 7) work better. Please note that the total number of parameters compared to S4 is drastically reduced on these tasks due to the smaller state size. Here is the result with 8 layers:
> > >
> > > **ListOps**
> > > | Model | Accuracy |
> > > |--- |---|
> > > |Liquid-S4 8 layers | 61.92 (0.2) |
> > > |Liquid-S4 9-layers | 62.75 (0.2) |
> > > | | p=5 |

---

> > > > ### Comment · Reviewer_WFdy · 2022-11-14
> > > > **Response**
> > > >
> > > > Thank you to the authors for their response.  I will respond to the most pertinent points, in order of most to least significant:
> > > >
> > > > ## Major
> > > >
> > > > **On discretisation**:  I would certainly expect to see expanded discussion of this.  You didn't mention anywhere in the paper that you require an Euler discretisation scheme.  Moreover, you say prior to (11, updated PDF), that you use a bilinear transform, which contradicts this.  If you are using the S4 code, are you using the bilinear-discretised parameters in the forward-Euler-discretised recurrence in (11, updated PDF)?
> > > >
> > > > **On PB autoregressive mode**:  I don't understand how PB can be used in autoregressive mode.  You write ''To obtain this, we set the transition matrix $\overline{A}$ in Liquid-S4 of Eq. 14, with an identity matrix, *only for the input correlation terms*.'' [emphasis added]  I read this as setting $\overline{A}$ in the purple terms in (12, updated PDF) to identity.  However, there is no way to then get back to the originally stated-discrete time recurrence in (11, updated PDF).  Similarly, I query if the wider theory and intuitions from the original LTC work carry across to the PB mode?
> > > >
> > > > **On reporting validation score**:  I don't believe that S4 [Gu+, 2022] report validation accuracy for LRA.  For certain ablations and exploratory experiments they report validation score (to avoid ``peeking'' at the test set).  Although generalisation error on these datasets is fairly small, the scores reported are still (IMO) incorrect.
> > > >
> > > > **On (b)**:  Thank you for including the KB results.  I would expect to see both variants included in the main text.  The LRA table has 25 other rows, and so an extra row making clear the difference between the two methods you present seems like a good use of space.
> > > >
> > > > **On feeling incremental**:  The authors should be proud of their work!  However, I am still unconvinced that the LTC ideas and theory carry across to the PB variant, how the LTC model has is discretised and parameterised, that (I still believe) fundamental aspects of S4 like the recurrent-mode have been lost.  The PB model appears to perform better across the board, and so really, the purest LTC formulation is not the best!  As a result of all of this, I view the method as adding an extra parameter dependency in to an existing model, and reporting better performance.  The only truly novel work here is showing that the liquid kernels can be represented using the same kernel structure, and that better performance can be achieved by using a more expressive model.
> > > >
> > > > ## Minor
> > > >
> > > > **On typographical errors**:  The paper is still littered with errors/inconsistencies, many of which I raised previously, and still haven't been fixed.
> > > > - Throughout:  Eq. X vs (X).
> > > > - Throughout: vectors should be boldfaced (cf. the notation used by Bishops PRML).
> > > > - Throughout: inconsistent capitalisation of ''L'' and ''S'' in ''Liquid-S4''.
> > > > - Below Eq. (3) (updated PDF):  ''... are input and output transition matrices...'' -> ''... are input and output matrices...''.
> > > > - Text above Eq. (5) (updated PDF): the mapping isn't from $u_k \rightarrow y_k$, it is from $u_{0:k} \rightarrow y_k$.
> > > > - Throughout:  Only proper nouns should be capitalised (eg. Normal Plus Low-Rank -> normal plus low-rank, Hippo Matrix -> HiPPO matrix).
> > > > - ''Computing S4 kernel Efficiently'' paragraph header:  ''kernel'' should be capitalised (all other headers are in title case).
> > > > - ''Computing S4 kernel efficiently'', Bullet Point 3:  $\mathbf{B}_{\mathbf{n}}$, the subscript $n$ should never be bolded, and when subscripted by $n$, the $B$ shouldn't be bolded when indexed by a single $n$ (cf. the notation used by Bishops PRML).
> > > > - ''Computing S4 kernel efficiently'', Bullet Point 5: Gu citation style is incorrect.
> > > > - $\mathbb{I}$ is much more common notation for the identity matrix.
> > > > - Bottom of Page 6:  Abbreviating to ''seq length'' isn't great.
> > > > - Table 1:  Variances are reported to one significant figure, whereas other methods have been reported to two decimal places.
> > > >
> > > > (this isn't even an exhaustive list.)
> > > >
> > > >
> > > > Thanks,
> > > >
> > > > WFdy

---

> > > > > ### Author Response · Authors · 2022-11-17
> > > > > **Response to New Response of Reviewer WFdy**
> > > > >
> > > > > Thank you very much for engaging in discussions with us.
> > > > >
> > > > > **On Discretization** We indeed use the bilinear transform form of A, B and C presented in (Eq. 3 updated) for the discrete weights of the system of (Eq. 11 updated). The clarification we provided in our response implies that the continuous system in (Eq. 10 updated) could be transformed directly to Eq. 11 by a forward Euler transformation. We further mentioned that due to the range of $\delta t$ and properties of A and B values, the bilinear transformed matrices presented in Eq. 3, would be close to the direct forward Euler system.
> > > > > We have revised this in our manuscript now, please see “Discretization of Liquid-SSMs” on Page 3.
> > > > >
> > > > > **On PB Autoregressive mode** In autoregressive (AR) mode, with PB (or any other conditioned kernel), we obtain $\overline{A}$, $\overline{B}$, and $\overline{C}$ no matter what the conditions are.
> > > > >
> > > > > More specifically, in the autoregressive mode of PB we can use Eq. 13. In Eq. 13, for computing the black parts we can reuse the AR mode of the plain S4 model and only have computed the new (violet) parts. As the violet parts consist of p multiplications of input terms (and the corresponding matrices) computing the AR mode is feasible. This adds a complexity of O(p) to the inference in the AR mode of PB, but because p is much smaller than L (past sequence length), it can significantly speed up the inference time compared to the convolution counterpart.
> > > > >
> > > > > Moreover, one of the properties of LTCs which was never studied and introduced before this work is their ability to account for the pairwise correlation of inputs which became apparent once we unrolled the system’s dynamics in this work. We believe that the pairwise correlation of inputs is a property that the PB kernel also possesses. Whether the kernel loses the expressivity and robustness attributes of LTCs, we have to investigate in future work.
> > > > >
> > > > > **On reporting Validation Score** We have no evidence in any of the S4 and S4D papers of a clear note on what they actually reporting in their papers. Please let us know if the reviewer knows of any evidence on this matter so that we update Table 1 with test accuracies. Nevertheless, to fully address the reviewer’s concern, we now reported accuracies on the test sets in the appendix.
> > > > >
> > > > > **On Kernel KB results** We have now included them in all tables.
> > > > >
> > > > > **On feeling incremental** We still believe that the reviewer could give more credit to our work and we are indeed very proud of our results that indicate how inductive biases in SSMs could improve the performance of this powerful class of models even further. Even if we limit the entire model to the KB kernel which directly reflects LTCs, the results on all benchmarks outperform S4 and all other S4D variants, let alone the PB kernel (given our clarifications) itself opens up exciting future research.
> > > > >
> > > > > Regardless of any change in decision, we would like to thank the reviewer very much for engaging in great discussions with us and for their amazing job as a reviewer providing feedback that actually improves our work.
> > > > >
> > > > > **Minors:** We are sorry that we missed some of these minor points in our revisions. We performed another round of revisions trying to address these minor issues which we greatly appreciate that are picked by the reviewer. However, please kindly note that this is not our camera-ready version, yet. We tried our best to address the raised major concerns and will do our best to address the rest of the typographical points (in case any of them remained) in the final version of the paper.
> > > > >
> > > > > Thank you,
> > > > >
> > > > > Authors

---

> > > > > > ### Author Response · Authors · 2022-11-26
> > > > > > **Feedback on the latest response**
> > > > > >
> > > > > > Dear Reviewer,
> > > > > >
> > > > > > We would like to inquire if our response addressed your latest points of concern. We really appreciate any feedback at this stage to clarify any remaining concerns, as we noticed that the reviewer did not change their evaluation score of our paper.
> > > > > >
> > > > > > Thank you,
> > > > > >
> > > > > > Authors.

---

> > > > > > > ### Comment · Reviewer_WFdy · 2022-11-26
> > > > > > > **Upgrading**
> > > > > > >
> > > > > > > To the authors,
> > > > > > >
> > > > > > > Thank you for your continued engagement.  In the short term, I am going to upgrade my score from a three to a five, and I will not advocate for this paper to be rejected.
> > > > > > >
> > > > > > > - I can see that the PB mode can be used in autogressive mode, but that all of the cross-terms have to be computed individually.  I think this is an exponential number of computations in $p$, at least looking at (12).  These products may be cheaper than the matrix-matrix products required for the naive S4 autoregression though, so maybe that is tolerable as well.  The authors should clarify this and comment on it in the text.
> > > > > > > - Thank you for the clarifications on the discretisation.  The arguments for discretisation seem a little hand wavy to me, but, whatever you do seems to work empirically, and so I will not press that any further here, but would expect a fairly detailed write-up _in the main text_ of whatever you actually implement, and any justification for it.
> > > > > > > - I find it really difficult that PB outperforms KB, and yet little dedicated analysis or discussion of the PB variant is included -- it is sort of "slipped in", when really it Is quite far removed from LTC models.  This should be highlighted as early on as in the abstract.
> > > > > > > - I am 95% confident that most other papers quote test scores for LRA.  If the authors are looking for more confirmation, I would reach out to, eg. the S4 authors to confirm what they did.  I would look to see the test table included in the main text.
> > > > > > > - I acknowledge that the authors have improved the flow and layout of the paper.
> > > > > > >
> > > > > > > Good luck,
> > > > > > >
> > > > > > > WFdy

---

> > > > > > > > ### Author Response · Authors · 2022-11-27
> > > > > > > > **Thank you very much**
> > > > > > > >
> > > > > > > > Thank you very much for upgrading your score and not advocating for the rejection of our work. We enjoyed our discussions very much as they were fair, constructive, and enlightening. Thank you for that.
> > > > > > > >
> > > > > > > > - As instructed, we will certainly elaborate more on the autoregressive mode of the PB kernel in the text.
> > > > > > > > - We will work on extending the description of our discretization scheme to shed more light on the empirical results and their formal justification in the main text.
> > > > > > > > - One possible reason why PB outperforms KB could be the fact that we limit the correlation terms with the truncation with order p. This limitation arises from how S4 blocks are constructed as a stack of many 1D blocks which does not computationally allow us to exploit the benefit of higher-order correlation terms due to the high computational complexity. This limitation might also reduce the expressivity of the KB kernel, but not PB, as they do not have a dependency on A for the correlation terms. A potential solution would be a Liquid-S5 instance, where we could directly use a parallel scan over the linear LTC system (which is a time-varying SSM). This is possible because we could precompute the state transitions at each time step. This way we would not need to truncate the kernel and obtain all correlation terms for free. This is an exciting extension to Liquid-S4 which we are exploring in future work.
> > > > > > > > - We have now asked the S4 authors to make sure the results are test accuracy and will update Table 1 accordingly.
> > > > > > > >
> > > > > > > > Thank you once again for helping us significantly improve our work.
> > > > > > > >
> > > > > > > > Authors

---

### Official Review · Reviewer_KBed · 2022-10-26

**Confidence:** 2
**Correctness:** 3
**Technical Novelty And Significance:** 4
**Empirical Novelty And Significance:** 4
**Recommendation:** 8

**Clarity, Quality, Novelty And Reproducibility:**

Well written.

There seems to be typo on page 6 .. .. "to use parametrization of $\Lambda - PP^*$ instead of $\Lambda - PP^*$"

**Strength And Weaknesses:**

Significant advance on the state of the art. I only have minor comments.

1. Why is the same B used for modifying the state transition matrix and additive input in Equation 8 ?
2. It is not clear why the best linearization of $A + Bu(t)$ equal to $\bar{A} + \bar{B}u(t)$. Ideally, the linearization must be done on $A + Bu(t)$ directly.
3. It seems that the empirical analysis only focuses on PB mode. Why? Since the paper presents two modes, it is best to present results with both modes in all tables (Liquid-S4-KB and Liquid-S4-PB).
4. It will be useful to get insights on the relationship between optimal $p$ and the "complexity" of $u$. Why does ListOps and IMDB require higher $p$ compared to Pathfinder and path-X datasets? Add Pathfinder and path-X to Figure 1 for completeness.

**Summary Of The Paper:**

The paper presents an extension on the popular S4 model by allowing the state transition matrix to be input dependent. They show significant improvements on the state of the art on many long range sequence prediction problems.

**Summary Of The Review:**

The paper presents a signficant advance. Minor changes can make the paper clearer.

---

> ### Author Response · Authors · 2022-11-13
> **Response to Reviewer KBed - Part 1**
>
> ## Part 1
>
> We would like to thank you very much for your positive evaluation of our work and your constructive feedback. In the following, we address your remaining comments and questions:
>
> **Question:** “Why is the same B used for modifying the state transition matrix and additive input in Equation 8 ?”
>
> **Answer** Choosing the same B as part of the autonomous system’s time constant 1/(A + Bu)  and input control creates a “coupling” effect in the LTCs’ original formulation (See Eq. 7), that appears to be an effective property for achieving better expressivity [1]. Inspired by the results presented in [1], we decided to preserve the structure of the LTC in the linearized version as well. Intuitively, selecting the same input transition matrix B simultaneously changes the characteristics (Time constant) of the LTC SSM and derives the input-dependent additive part of the ODE, therefore, establishing the coupling behavior.
>
> [1] Hasani et al. AAAI 2021
>
> **Question:** “It is not clear why the best linearization of $A+Bu(t)$ equal to $\overline{A}+\overline{B} u(t)$. Ideally, the linearization must be done on A+Bu(t) directly.”
>
> **Answer:** The dynamical system presented in Eq. 8, is a continuous-time (CT) bilinear state-space model (SSM). Ideally, we want that the discretization of a CT bilinear SSM 1) satisfy the first-order form of the model, and 2) preserve the bilinear model structure. This is challenging and only possible via a limited number of methods:
>
> 1) The most straightforward approach is to use Forward Euler with first-order error:
> $\dot{x} = \frac{x_{k+1} - x_{k} }{\delta t} + \mathcal{O}(\delta t)$. Now by plugging this in Eq. 8, we get $\overline{A} = I + \delta t A$, $\overline{B} = \delta t B $. Such discretization satisfies the conditions above. For this discretization to stay stable, for $s=\sigma \pm i \omega$ an eigenvalue of the continuous transition matrix A, and $\lambda = 1 + s \delta t $, an eigenvalue of the discrete model, $\mathcal{R}e(s) \leq 0$ or $|\lambda| = |1 + s \delta t | \leq 1$, thus $(1 + \sigma \delta t)^2 + \omega^2 \delta t^2 \leq 1$. This condition implies that selecting a small enough $\delta t$ ensures the system's stability, but for cases where $\delta t$ is large, the system might go unstable.
>
> We choose this straightforward Forward Euler discretization. We can show that based on the properties of the transition matrices A and B, and the range of selected $\delta t$ a bilinear transformation of discrete matrices $\overline{A}$ and $\overline{B}$, would be very close to that of our Forward Euler discretization. This means that:
>
> $ |  \overline{A} _{\text{Forward Euler}} - \overline{A} _{\text{Approx bilinear}} | < \gamma ~~~~ 0 < \gamma < \frac{delta t}{2} $
>
> 2) Adams-Bashforth Method: The second-order Adams-Bashforth will apply the transformation $x_{k+1} = x_k + \frac{3 \delta t}{2} f(k) - \frac{\delta t}{2} f(k-1) + \mathcal{O}(\delta t^2)$, where f(k) is the right-hand-side of Eq. 8 at time $t= k \delta t$. This method also satisfies the two conditions we required.
>
> Please denote that computing a bilinear transform (https://en.wikipedia.org/wiki/Bilinear_transform) of a continuous-time bilinear SSM while preserving the first-order structure of the model is an open problem. As you mentioned, ideally, we can apply this transformation on A+Bu(t). However, it is challenging to preserve the first-order form of the equation while keeping the bilinear (liquid) structure of the model described in Eq. 8. We will certainly add an appendix and import this discussion to our revised version.

---

> > ### Author Response · Authors · 2022-11-13
> > **Response to Reviewer KBed - Part 2**
> >
> > ## Part 2
> >
> > **Question:** “empirical analysis only focuses on PB mode. Why? Since the paper presents two modes, it is best to present results with both modes in all tables (Liquid-S4-KB and Liquid-S4-PB).”
> >
> > **Answer** As we stated on Page 16 of our submission: We choose our simplified PB kernel over the KB kernel due to the computational costs and performance. Liquid-S4’s PB kernel is faster and on average provides better performance. That is the reason why we chose not to dilute the information in tables with various variants and instead consistently provide results for the PB kernel. However, if the reviewer still thinks that the results for the KB kernel are necessary, we will certainly include them either in the main tables or as an appendix. Please let us know what you think.
> >
> > **Question:** “It will be useful to get insights on the relationship between optimal p and the "complexity” of u. Why does ListOps and IMDB require higher p compared to Pathfinder and path-X datasets? Add Pathfinder and path-X to Figure 1 for completeness.”
> >
> > **Answer** This is a fantastic question. Indeed, we empirically observed that accounting for higher order correlations (i.e., higher liquidity order, p) for almost all datasets enhances performance except for the case of the speech command recognition task. Why this is the case, we do not know yet, and will certainly be investigating it in future work.
> >
> > For Path-X, we have not tested higher p values due to compute limitations (please see the compute time required in our response to reviewer xtka). However, as requested by the reviewer, we tested higher orders for other datasets which we presented in Figure 2 in our revised manuscript. For the pathfinder dataset, we see a plateau in performance as we increase the order.

---

### Official Review · Reviewer_xtka · 2022-10-28

**Confidence:** 4
**Clarity, Quality, Novelty And Reproducibility:** The paper is easy to read and present…
**Correctness:** 3
**Technical Novelty And Significance:** 3
**Empirical Novelty And Significance:** 4
**Recommendation:** 6

**Strength And Weaknesses:**

Strengths:
- Input dependent transition mechanism in state space models is a novel, important and natural extention of the original S4 model. I really like that this paper thoroughly explores it.
- The paper conducts thorough experiments on multiple datasets and get impressive improvements over the original S4 model.

Weaknesses:
The main set of weaknesses I see in this paper is related to computational complexity and cost.

- This paper can be much stronger with a simpler diagonal version of Liquid-S4. Some of the accelerators don't support (or are super expensive) the full set of operations required for implementing S4.
- The paper also doesn't report relative increase in computational cost or wall clock time compared to S4 and other alternatives. I would be interested in knowing that.
- [Optional] Some of the the baselines reported on this paper on LRA doesn't match what's reported in the original papers e.g. S5 paper. Although, its possible that they were updated recently. It's likely that some of the improvements are orthogonal can improve results of this paper and make it even stronger.

**Summary Of The Paper:**

This paper generalizes the recently proposed S4 model with an input dependent time constant used in state transition matrix and conduct thorough experiments to illustrate its benefits.

**Summary Of The Review:**

While the paper introduces very interesting ideas and impressive results, it can be made much stronger by also looking at computational costs attached to introducing these ideas.

---

> ### Author Response · Authors · 2022-11-13
> **Response to Reviewer xtka**
>
> We would like to thank the reviewer very much for their positive review of our work and their constructive feedback on our manuscript. In the following we address their remaining concerns to hopefully motivate a clear acceptance score:
>
> **Question:** “This paper can be much stronger with a simpler diagonal version of Liquid-S4. Some of the accelerators don't support (or are super expensive) the full set of operations required for implementing S4.”
>
> **Answer:** We fully agree and have already implemented a Liquid-S4 Diagonal kernel which has been submitted with our code at the time of submission. This kernel can be called by the following flags:
>
> ```python3 train.py wandb=null experiment=ml/cifar model.layer.liquid_kernel=polyb model.layer.mode=diag```
>
> Similar to S4 vs S4D kernels, there is a performance drop with the diagonal kernels, but the models are faster to run. We hope that this response addresses the reviewer’s concern on the accessibility of the Liquid-S4 kernels on various accelerators that do not support the full set of operations required for implementing S4-LegS.
> **Question:** “The paper also doesn't report relative increase in computational cost or wall clock time compared to S4 and other alternatives. I would be interested in knowing that.”
> **Answer:** We provided the time complexity analysis of Liquid-S4 in Page 6 under “Computational Complexity of the Liquid-S4 Kernel”. To address the reviewer’s feedback, we computed the time-per epoch for S4 and Liquid-S4 with various orders p and reported them below for all tasks in the Long Range Arena benchmark.
>
> **Time Per epoch (min)**
> |Dataset | IMDB | ListOps | sCIFAR | AAN | Pathfinder | Path-x |
> |---|---|---|---|---|---|---|
> | S4 - LegS | 3.49 | 4.60 | 1.59 | 45.40 | 7.31 | 143.93 |
> | Liquid-S4 p= 2 | 3.87 | 5.56 | 2.16 | 54.67 | 9.85  | 183.26 |
> | Liquid-S4 p= 3 | 4.10 | 6.53  | 2.70 | 64.21 | 11.92  | 221.71 |
>
> **Question:** [Optional] Some of the the baselines reported on this paper on LRA doesn't match what's reported in the original papers e.g. S5 paper. Although, its possible that they were updated recently. It's likely that some of the improvements are orthogonal can improve results of this paper and make it even stronger.
>
> **Answer:** your intuition is 100% correct. We have included the results for S5 from the first version of the preprint released on August 9th, 2022 in the following link: https://arxiv.org/pdf/2208.04933v1.pdf
>
> In our revised manuscript, we will definitely add the updated results of the S5 paper from their latest preprint update which was released on Oct 6th, 2022 (After ICLR’s submission deadline) here: https://arxiv.org/abs/2208.04933v2
>
> We agree and will certainly investigate the orthogonalization of liquid-S4 kernels in a more structured way in future work. We have also been entertaining the idea of implementing a potential Liquid-S5 version in the near future where the elegant parallelization scheme introduced in the S5 paper could potentially improve the efficiency and performance of liquid-S4 even further.

---

> > ### Author Response · Authors · 2022-11-30
> > **Follow up with Reviewer xtka**
> >
> > Dear Reviewer xtka,
> >
> > We provided a detailed response to your comments and performed the experiments you requested on providing the relative increase in computational costs of our method as well as diagonal versions of Liquid-S4.
> >
> > We would love to hear back from you and make sure that all your concerns are addressed and hopefully motivate a higher review score.
> >
> > Thank you very much,
> >
> > Sincerely,
> >
> > Authors

---

### Official Review · Reviewer_qgqj · 2022-10-29

**Confidence:** 2
**Correctness:** 3
**Technical Novelty And Significance:** 3
**Empirical Novelty And Significance:** 3
**Recommendation:** 8

**Clarity, Quality, Novelty And Reproducibility:**

The paper is written in a clear and precise scientific language. The proposed method is novel in its own context, even though probably not as novel for the signal processing domain. The paper gives sufficient amount of details to reproduce the experiments.

**Strength And Weaknesses:**

Strengths:
i) The proposed method is very comprehensive. It is tailored in fine details including computational feasibility.
ii) The paper reports a large list of experiments, compares against competitive modern machine learning approaches such as transformers, and appears to outperform all of them.

Weaknesses:
i) The method involves calculation of iFFT somewhere in the pipeline. I wonder how much it could preserve end-to-end differentiability of the eventual loss function if it is used together with encoders, for instance in high-dimensional sequences such as videos.

**Summary Of The Paper:**

The paper introduces an approach for capturing long-term dependencies of complex time series. The paper incorporates some ideas from the classical dynamical system modeling literature into machine learning.

**Summary Of The Review:**

Solid piece of work. The approach is a bit different from the commonplace approaches of the core machine learning community, which could make its accessibility a bit questionable. On the other hand, this could also be an opportunity for the community to get exposed to ideas from different domains.

---

> ### Author Response · Authors · 2022-11-13
> **Response to Reviewer qgqj**
>
> We would like to thank you very much for your positive evaluation of our work. In the following, we will address your remaining concern:
>
> **Question:** “The method involves calculation of iFFT somewhere in the pipeline. I wonder how much it could preserve end-to-end differentiability of the eventual loss function if it is used together with encoders, for instance in high-dimensional sequences such as videos”
>
> **Answer:** iFFT is differentiable and can be used with autograd tools in a straightforward manner in any architecture [1, 2]. Therefore there is no issue to use Liquid-S4 blocks as drop and place modules together with encoders. Moreover, Similar to S4, Liquid-S4 is also built by stacking 1-D state-space models (SSMs) that perform very well on sequential data. As the dimensionality of input sequences goes higher (e.g., videos) it makes sense to use Liquid-S4 as a blackbox module to build N-dimensional versions of it to use for video data. This is shown in a very recent work [3] with N-dimensional arrays of S4 blocks which could be directly performed by Liquid-S4 blocks as well.
>
> [1]https://pytorch.org/blog/the-torch.fft-module-accelerated-fast-fourier-transforms-with-autograd-in-pyTorch/
>
> [2] https://arxiv.org/pdf/2002.02709.pdf
>
> [3] Nguyen et al. NeurIPS 2022, S4ND: Modeling Images and Videos as Multidimensional Signals Using State Spaces https://arxiv.org/pdf/2210.06583.pdf

---

> > ### Comment · Reviewer_qgqj · 2022-11-28
> > **Keep my grade**
> >
> > Thanks for your response. My concerns have been addressed.

---

> > > ### Author Response · Authors · 2022-11-30
> > > **Thank you very much**
> > >
> > > Dear Reviewer qgqj,
> > >
> > > Thank you very much for voting for the acceptance of our paper and for your valuable comments and questions.
> > >
> > > Sincerely,
> > >
> > > Authors

---

### Decision · Program_Chairs · 2023-01-20

**Decision:**

Accept: poster

**Justification For Why Not Higher Score:**

see above

**Justification For Why Not Lower Score:**

Indeed, I was inclined to go for a reject at first.  Rebuttal of the authors was convincing though, and the final paper will get enough attention to merit publication.

**Metareview: Summary, Strengths And Weaknesses:**

The paper extends the S4 model by allowing the state transition to be input-dependent, thus improving results on long sequences.

On the positive side, the paper is supported by a large number of experimental results.

On the downside, one can turn that around and say the paper is incremental and heavily experimental.  It does not add to science in a big way.



**Note From Pc:**

if the above contains the word "oral" or "spotlight" please see: "oral" presentation means -> notable-top-5% and "spotlight" means -> notable-top-25%. As stated in our emails, we are disassociating presentation type from AC recommendations

**Summary Of Ac-Reviewer Meeting:**

The planned AC meeting ended in a reviewer changing their mind and the paper merited an accept.